# Aging explains earlier start of leaf senescence in trees during warmer years: translating the latest findings on senescence regulation into the DP3 model (v1.1)

5   Michael Meier*[1], Christof Bigler[2], Isabelle Chuine[1]

[1] CEFE, Univ Montpellier, CNRS, EPHE, IRD, Montpellier, France

[2] Forest Ecology, Department of Environmental Systems Science, ETH Zurich, Zurich, Switzerland

*Correspondence to*: Michael Meier, (michael.meier@cefe.cnrs.fr)

**Short summary.** The DP3 model of leaf coloring, formulated according to the leaf development process, considerably contrasts previous models and allows to set up new hypotheses, e.g., regarding earlier senescence induction and longer senescence predicted for warmer conditions. Comparing the accuracy of the DP3 model to that of previous models and the Null model (average observed date of leaf coloring) suggests that leaf coloring data are noisy, which is why obser-

vation protocols and methods should be revised.

**Abstract.** Leaf senescence ends the growing season of deciduous trees, affecting the amount of atmospheric $CO_2$ sequestered by forests. Therefore, some climate models integrate projected leaf senescence dates to simulate the carbon cycle. Here, we developed a process-oriented model of leaf senescence (the 'DP3 model') by testing 34 formulations of

the leaf development process based on the latest findings on the regulation of leaf aging and senescence. The period between leaf unfolding and leaf senescence was separated into the subsequent young, mature, and old leaf phases, with particular reactions to leaf aging and cold stress, photoperiod stress, and dry stress. The DP3 model predicts daily rates of aging and stress (i.e., separated in cold, photoperiod, and dry stress) along with dates of transition from young to mature to old leaf, senescence induction dates, and leaf senescence dates. This allows new hypotheses regarding the regula-

tion of leaf senescence to be tested. For example, the DP3 model predicted earlier senescence induction in warmer conditions due to aging and together with longer senescence, which can be validated with experiments and in situ observations. The DP3 model and compared previous models were equally accurate, but less accurate than the Null model (average senescence date observed in the calibration sample). This lower accuracy of the DP3 and compared models is likely due to noise in the visually observed leaf senescence data, which blurs the signal of the leaf senescence process,

and to incorrect model formulations. The model errors were similarly affected by climate conditions and location among compared models (including the Null model) and varied mostly due to the leaf senescence data. Noisy leaf senescence data likely force the models to resort to the mean observation, impeding inferences from accuracy-based model comparisons about the leaf senescence process. This calls for revised observation protocols and methods that measure rather than estimate different senescence stages, such as senescence induction and 50% of the leaves have

changed color, e.g., based on greenness, involving digital cameras and automated image assessment.

**Keywords:** Leaf phenology, process-based model, senescence induction, senescence onset, leaf coloring, leaf fall, data, observation protocols, observer bias, sample bias, bias towards the mean

## 1        Introduction

Leaf senescence involves several processes and regulation pathways, but the most important process is the degradation
of chlorophyll and breakdown of chloroplasts to retrieve nutrients, especially nitrogen, and to mobilize them in new
leaves in spring (Cooke and Weih, 2005; Keskitalo et al., 2005; Lim et al., 2007; Rogers, 2017). A side effect of this nu-
trient resorption is the change in leaf color from green to yellow, orange, or red (Keskitalo et al., 2005; but see Wheeler
and Dietze, 2023). There have been many studies of how the timing of leaf coloring is influenced by climatic conditions
(e.g., Bigler and Vitasse, 2021; Liu et al., 2018; Meier et al., 2021). As these studies usually used the term 'leaf color-
ing' or 'leaf senescence' to refer to a particular stage of leaf senescence, we use 'leaf senescence' as a collective term for
the stages when a given relative amount of leaves have changed color or have fallen, unless stated otherwise.

Leaf senescence marks the end of a process that has been better understood over the last ten years, mainly
thanks to studies in cell and molecular biology and in environmental sciences (Fig. 1). These studies have shown that
leaf senescence relates to leaf development state (e.g., Jan et al., 2019; Jibran et al., 2013; Lim et al., 2007). On the one
hand, the development state of leaves depends on their age and thus on the time since leaf unfolding and the state of car-
bohydrate sinks (Jibran et al., 2013), which relates to photosynthetic activity and nutrient availability (Paul and Foyer,
2001). While earlier leaf unfolding was related to earlier leaf senescence (Fu et al., 2014, 2019), an intense discussion
has started about the possibility of earlier leaf senescence due to increased photosynthetic activity (Kloos et al., 2024;
Lu and Keenan, 2022; Marqués et al., 2023; Norby, 2021; Zohner et al., 2023). On the other hand, the development
state of leaves is influenced by hormone levels (Addicott, 1968; Jan et al., 2019; Jibran et al., 2013; Lim et al., 2007),
which are, among others, stimulated by environmental stress caused by cold (Kloos et al., 2024; Wang et al., 2022; Xie
et al., 2015, 2018), drought (Bigler and Vitasse, 2021; Mariën et al., 2021; Tan et al., 2023; but see Kloos et al., 2024;
Xie et al., 2015, 2018), heat (Bigler and Vitasse, 2021; Mariën et al., 2021; Tan et al., 2023; Xie et al., 2015, 2018),
heavy rain (Kloos et al., 2024; Xie et al., 2015, 2018), short days (Addicott, 1968; Keskitalo et al., 2005; Singh et al.,
2017; Tan et al., 2023; Wang et al., 2022), and lack of nutrients (Fu et al., 2019; Tan et al., 2023). In the early phase of
leaf development ('young leaf'), senescence cannot be induced, whereas aging and stress induce it in later phases ('ma-
ture leaf' and 'old leaf') and regulate the rate of senescence (Jan et al., 2019; Jibran et al., 2013; Lim et al., 2007; Paul
and Foyer, 2001; Tan et al., 2023).

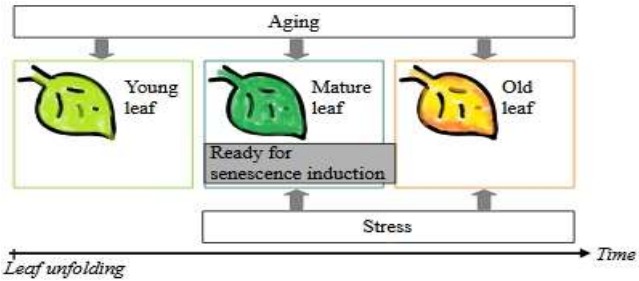

**Figure 1. Leaf development.** Starting with leaf unfolding, the young leaf develops first into a mature leaf and then into an old leaf.
During the three phases of the young, mature, and old leaf, the leaf ages continuously. With the transition from young to mature leaf,

the leaf becomes ready for senescence induction through environmental stress (e.g., cold days). If senescence is not induced through stress in the mature leaf, it certainly is through aging with the transition from mature to old leaf. Thus, senescence cannot be induced during the phase of the young leaf and the date of senescence induction depends on the time since leaf unfolding and the environmental conditions since the transition from young to mature leaf. Adapted from Figure 1 in Jibran et al. (2013).

As the timing of senescence induction depends on environmental conditions, leaf senescence of deciduous trees shifts as climate changes, which influences the timing and length of their growing season and thus affects the amount of $CO_2$ absorbed from the atmosphere (Meier et al., 2021; Menzel et al., 2020; Piao et al., 2019; but see Mariën et al., 2021). This links the feedback loop between atmospheric $CO_2$ concentration and climate to the feedback loop between climate and forests and more generally to terrestrial ecosystems (Luo, 2007; Richardson et al., 2013). Further, the

amount of absorbed $CO_2$ relates to the amount of sugars available for tree growth, defense, and reproduction (Herms and Mattson, 1992; Tan et al., 2023). Therefore, accurate projections of leaf senescence dates under a changing climate are necessary for accurate forecasts of both climate change and future species composition of temperate forests.

      Leaf senescence dates are often projected using process-oriented models. These models are usually based on the results of experiments testing the effect of various environmental cues, that are translated mathematically (Chuine et

al., 2013; Chuine and Régnière, 2017). Various process-oriented models of leaf senescence have been proposed over the last twenty years (Liu et al., 2020; Meier and Bigler, 2023). They generally formulate leaf senescence as a one-way process that starts shortly after summer solstice by accumulating a daily rate of senescence until a threshold is reached (but see Wheeler and Dietze, 2023). The daily rate is usually dependent on temperature and day length, and the threshold is either a constant or depends on the timing of leaf unfolding, or on temperature, precipitation, and photosynthetic

activity during the growing season (e.g., Delpierre et al., 2009; Keenan and Richardson, 2015; Liu et al., 2019; Zani et al., 2020).

      Previous studies have shown that these leaf senescence models are heavily biased towards the mean of the calibration sample (Meier et al., 2023) and are less efficient relatively to leaf unfolding models (e.g., Liu et al., 2020; Meier and Bigler, 2023). However, it is not yet clear whether this is due to noisy phenological data and/or an incomplete

process formulation.

      The phenological data used to train leaf senescence models have often been recordings of visual observations, which cover long time periods and are species-specific (e.g., ongoing since 1951 in the Swiss phenology network, 2025). However, the observations are noisy due to different observers and small sample sizes. For leaf senescence, Liu et al. (2021) showed for example that the observer bias was 15 days [d] (median) and the sampling bias was 10 d (me-

90 dian) for 10 trees observed per population. These biases not only lead to noise between sites, but also within sites when observers and samples change. Such changes can lead to breaks in the time series, as was found for some Swiss sites (Auchmann et al., 2018; Swiss phenology network, 2025). Moreover, the observation protocols may differ between the meteorological institutes and citizen science based networks that are responsible for the recording in the different European countries (Menzel, 2013).

Current models formulate leaf senescence as the result of an accumulated stress caused by cold and short days after summer solstice (Delpierre et al., 2009; Dufrêne et al., 2005; Keenan and Richardson, 2015; Lang et al., 2019; Liu et al., 2019; Zani et al., 2020). Two models further consider environmental conditions before summer solstice, either through temperature and precipitation during the growing season (Liu et al., 2019) or through the photosynthetic activity during the growing season (Zani et al., 2020), while one model considers age through the timing of leaf

unfolding (Keenan and Richardson, 2015). However, in these models, environmental conditions and age affect the amount of stress needed for leaf senescence rather than senescence induction. In other words, according to current models, the senescence induction depends only on stress caused by cold and short days, which considerably contrasts with current knowledge (see above; Fig. 1). None of the current models allows for senescence induction caused by aging. While two models consider stress that occurred before summer solstice, senescence is always induced after summer solstice. Finally, we are unaware if aging, stress caused by other than cold and short days, and different stress effects among the phases of leaf development have been tested, as none of the corresponding studies mentioned tested but discarded model formulations.

Here, we developed a new process-oriented model that simulates the timing of leaf senescence based on the latest knowledge of the physiological processes and drivers of leaf senescence. The timing of leaf senescence was formulated through a leaf development process that starts at leaf unfolding and is driven by aging and various types of abiotic stress. We tested 34 model formulations of this process. Finally, the most accurate formulation was evaluated with a particular focus on the differences between the simulated and observed values (i.e., 'model errors'). We addressed the following research questions:

(1) Which model formulation yields the most accurate simulations of the timing of leaf senescence?

(2) How accurately does this model simulate leaf senescence compared to previous models?

(3) How do the model errors relate to the phenological data, climate, and site conditions?

## 2 Data and methods

### 2.1 Phenological data

The model was developed and evaluated with leaf phenology data of common beech (*Fagus sylvatica* L.), which was visually observed in Austria, Germany, Switzerland, and Great Britain between 1950 and 2022 (Fig. 2, Table 1, Sect. S1.1; PEP725, 2024; Swiss phenology network, 2025; Templ et al., 2018). We used the phenological stages 50% of the leaves are unfolded as well as 50% and 100% of the leaves have changed color or have fallen (hereafter referred to as 'leaf unfolding' [LU], 'leaf senescence$_{50}$' [LS50], and 'leaf senescence$_{100}$' [LS100], respectively; corresponding to BBCH15, BBCH95, and BBCH97 according to Meier, 2018). The LS$_{100}$ data were recorded in Austria and Great Britain only.

We checked all site-years with regards to the order and completeness of the phenological observations. Observations of LS$_{50}$ and LS$_{100}$ that occurred between the day of year (doy) 60 and 151 were discarded, as were observations of LU that occurred after doy 180 or after LS$_{50}$ or LS$_{100}$. Thus, we considered only site-years with an observation for LU that was followed by either LS$_{50}$ or LS$_{100}$, or by both LS$_{50}$ and later LS$_{100}$, leaving 5018 sites.

From these sites, we made a pre-selection so that the phenological and geographical range of the LS$_{50}$ observations was evenly covered and all LS$_{100}$ observations were included. This involved splitting all 5018 sites into 8–10 bins with equal spans for the average and standard deviation of LS$_{50}$ as well as for latitude, longitude, and elevation, so that each bin contained at least two sites (e.g., the range between doy 232 and 328 for the average LS$_{50}$ was split into ten bins of 9.7 days). From each bin, we chose the site with the most LS$_{50}$ observations, with random choice if this applied to more than one site. These sites were completed by all sites with an LS$_{100}$ observation, resulting in a pre-selection of 7137 LS$_{50}$ and 850 LS$_{100}$ observations recorded at 244 and 106 sites, respectively.

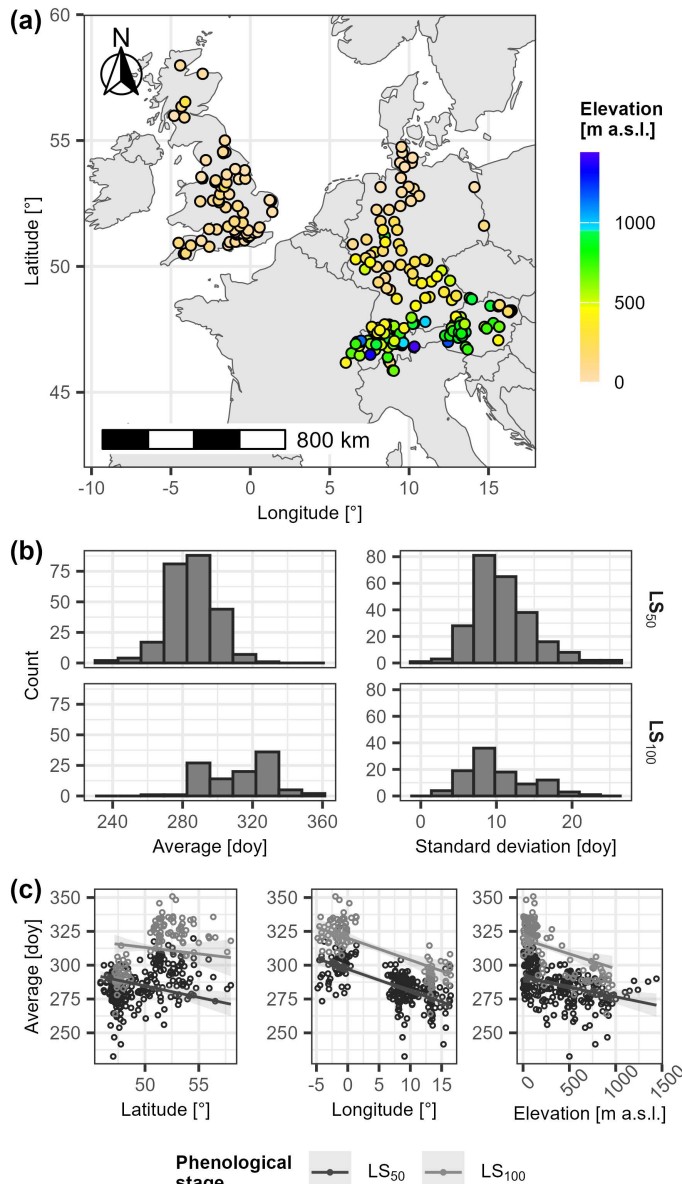

**Figure 2. Selected phenological sites.** Panel (a) locates the selected sites and indicates corresponding elevation [meters above sea level (m a.s.l.)]. In (b), the histograms illustrate the distributions of the site-specific average day of year (left) and corresponding standard deviation (right) per leaf senescence stage (i.e., 50% and 100% of the leaves have changed color or have fallen [$LS_{50}$ and $LS_{100}$, respectively]; rows). Panel (c) plots the site-specific average day of year of $LS_{50}$ and $LS_{100}$ (grey and black circles, respectively) in relation to site latitude [°] (left), longitude [°] (middle), and elevation [m a.s.l.] (right), together with the linear regression lines and corresponding 99% confidence intervals. Site-specific average $LS_{50}$ and average $LS_{100}$ were explained by latitude, longitude, and elevation (Sect. S1.1.2) and corresponding estimates were plotted against latitude, mean longitude and mean elevation (left), longitude, mean latitude, and mean elevation (middle), and elevation, mean latitude, and mean longitude.

**Table 1. Observations of spring and autumn leaf phenology.**

| Stage | Country | Sites | Total number of site-years (min.–max. per site) | Observation period | Range of observations [doy] | Source |
|---|---|---|---|---|---|---|
| $LS_{50}$ | Austria | 51 | 1011 (5–54) | 1950–2015 | 209–321 | PEP725 |
| | Germany | 68 | 3238 (14–65) | 1951–2015 | 196–331 | PEP725 |
| | Great Britain | 64 | 303 (2–6) | 1999–2005 | 258–337 | PEP725 |
| | Switzerland | 61 | 2585 (6–72) | 1951–2022 | 197–344 | SPN |
| $LS_{100}$ | Austria | 43 | 578 (1–34) | 1950–1986 | 263–335 | PEP725 |
| | Great Britain | 63 | 272 (1–6) | 1999–2005 | 286–365 | PEP725 |

| LU | | | | | | |
|---|---|---|---|---|---|---|
| | Austria | 51 | 1020 (5–54) | 1950–2015 | 80–166 | PEP725 |
| | Germany | 68 | 3238 (14–65) | 1951–2015 | 80–175 | PEP725 |
| | Great Britain | 64 | 331 (5–6) | 1999–2005 | 85–140 | PEP725 |
| | Switzerland | 61 | 2585 (6–72) | 1951–2022 | 67–161 | SPN |

*Note*: LU refers to the stage when 50% of the leaves are unfolded. $LS_{50}$ and $LS_{100}$ refer to the stages when 50% and 100% of the leaves, respectively, have changed color or have fallen. The timing of these stages is given by the day of year (doy). A site-year is a year for which an observation of both LU and $LS_{50}$ or $LS_{100}$ was recorded at a given site. Two data sources were considered: PEP725 (Templ et al., 2018) and the Swiss phenological network (SPN; Swiss phenology network, 2025).

## 2.2 Driver data

For each phenological site, weather variables, elevation, and the leaf area index (LAI) were approximated by the weighted averages from octagons with a radius of 2.5 km around the phenological sites, and combined with the atmospheric $CO_2$ concentration. Daily weather variables and elevation were derived for each site from the E-OBS dataset (Copernicus Climate Change Service, Climate Data Store, 2020; Cornes et al., 2018), which contains interpolated data from a 100-member ensemble driven with meteorological observations. We extracted and approximated site elevation, maximum temperature, mean temperature, minimum temperature, precipitation, relative humidity, and surface shortwave down welling radiation for 1950–2022. These temperature variables were corrected through day- and site-specific lapse rates to account for elevational differences between the octagon averages and sites (i.e., the elevation according to the phenology datasets or, if missing, according to EU-DEM, 2024, with a resolution of 25 m, and the location according to the phenology datasets). These laps rates were linearly regressed from the grid cell of a particular site and the eight neighboring grid cells, assuming an elevation of 0 meters above sea level [m a.s.l.] for grid cells over the sea. Occasional gaps in the regressed lapse rates were interpolated with site-specific cubic splines. LAI per site was taken from the remote sensed monthly LAI (1981–2015) in the GIMMS-LAI3g dataset (version 2; Mao and Yan, 2019). LAI is averaged among years in this dataset, and thus we also used these monthly LAI values for the years 1950–1980. Atmospheric $CO_2$ concentrations were taken from a reconstructed and a remote sensed dataset for the years 1950–2013 and 2002–2022, respectively (Cheng et al., 2022; Copernicus Climate Change Service, Climate Data Store, 2018). Both datasets provide monthly data, which we distilled into annual averages. These averages were combined through weighted means over the years 2002–2013 to assure a smooth transition between the datasets. As some monthly $CO_2$ observations between 2002–2022 were missing, we used modeled $CO_2$ values derived from site-specific cubic splines based on the remote sensed data (Copernicus Climate Change Service, Climate Data Store, 2018).

We further calculated for each site day length, daily photosynthetic activity, and the daily Keetch and Byram drought index (KBDI). Day length was calculated following Brock (1981), using the latitude of each site (Sect. S1.2.1). Daily sink limited photosynthetic activity was calculated following Farquhar et al. (1980) and Collatz et al. (1991), using daily surface shortwave down welling radiation, day length, and mean temperature together with monthly atmospheric $CO_2$ concentration as well as monthly LAI averaged among years (Sect. S1.2.2). The daily KBDI was calculated following Keetch and Byram (1968), using precipitation and maximum temperature (Sect. S1.2.3).

## 2.3 Model conceptualization

Based on the process of leaf development according to Jibran et al. (2013), we defined our model as a one-way process that may be formulated with either two or three phases of leaf development, namely either the phases mature and old leaf or the phases young, mature, and old leaf (Figs. 1 and 3). After leaf unfolding, the young leaf is insensitive to stress

and ages until it becomes a mature leaf (Fu et al., 2014; Jibran et al., 2013; Keenan and Richardson, 2015). The mature leaf can be affected by stress and ages until it becomes an old leaf (Jan et al., 2019; Jibran et al., 2013; Lim et al., 2007). Senescence may be induced by stress in the mature leaf or by aging, in the case of which it coincides with the transition from mature to old leaf, causing the leaf to change color and to fall off (Jan et al., 2019; Jibran et al., 2013; Lim et al., 2007).

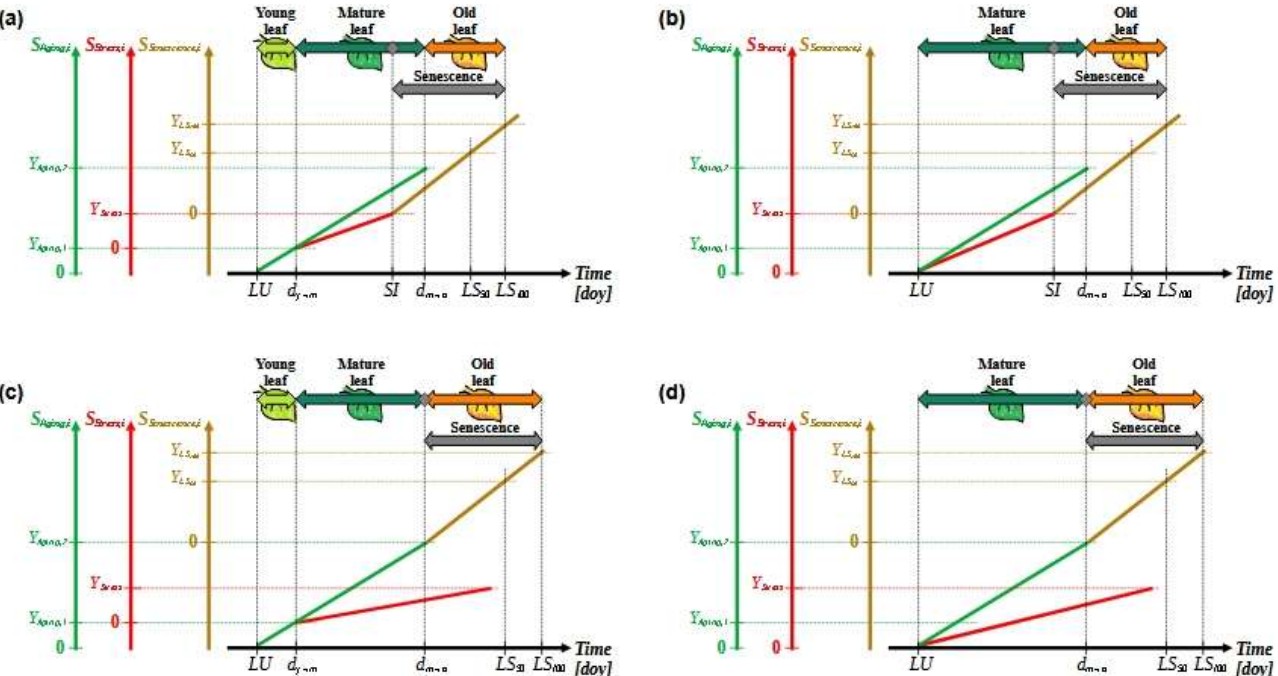

**Figure 3. Conceptualization of the leaf development model.** The process of leaf development is defined by three subsequent phases of leaf development, i.e., 'young leaf', 'mature leaf', and 'old leaf' [light green, dark green, and orange horizontal arrows, respectively; panel (a) and (c)]. Alternatively, the process is simplified into two subsequent development phases, i.e., 'mature leaf', and 'old leaf' (panel b and d). Senescence may be induced by stress during the phase of the mature leaf [grey rhombuses; panels (a) and (b)] or by aging on the day of transition from the mature to the old leaf [panels (c) and (d), respectively]. The state of aging, stress, and senescence (y-axes; $S_{Aging,i}$, $S_{Stress,i}$, and $S_{Senescence,i}$; Eq. 1; solid green, red, and brown lines, respectively) for day $i$ are derived from the corresponding daily rates (Eqs. 3, 4, and 8) accumulated over time (x-axis; day of year [$doy$]). Starting from the day of leaf unfolding ($LU$), these states simulate the leaf development, marked by transitions from the young to the mature leaf ($d_{y \to m}$) and from the mature to the old leaf ($d_{m \to o}$) as well as by the dates of senescence induction ($SI$) and of the phenological stages 50% and 100% leaf coloring or fall ($LS_{50}$ and $LS_{100}$, respectively). These transitions and stages occur when $S_{Aging,i}$, $S_{Stress,i}$, and $S_{Senescence,i}$ breach corresponding thresholds ($Y_{Aging,1}$, $Y_{Aging,2}$, $Y_{Stress}$, $Y_{LS50}$, and $Y_{LS100}$). $SI$ is defined as the first day on which either $Y_{Stress}$ or $Y_{Aging,2}$ is breached [panels (a) and (b) versus panels (c) and (d), respectively] and marks the beginning of senescence (grey horizontal arrow), during which the daily senescence rate accumulates. If $SI$ results from the breach of $Y_{Aging,2}$, it coincides with $d_{m \to o}$. Dotted lines are auxiliary lines.

We constructed and tested the formulations of leaf development (see Sect. 2.2.3) by combining the following assumptions. We considered that aging could be modeled either by photosynthetic activity (Jibran et al., 2013; Paul and Foyer, 2001; Zohner et al., 2023) or more simply by a number of days. Stress may be modeled by a combination of the stressors cold, shortening day length, drought, heat, frost, heavy rain, and nutrient depletion (Bigler and Vitasse, 2021; Jan et al., 2019; Jibran et al., 2013; Kloos et al., 2024; Mariën et al., 2021; Tan et al., 2023; Wang et al., 2022; Xie et al., 2015, 2018; Zohner et al., 2023). Finally, we considered that leaf senescence could result as a combination of aging and stress (Tan et al., 2023; Xie et al., 2015).

All formulations are based on daily states of aging, stress, and senescence (Eq. 1), which are compared to corresponding thresholds (Eq. 2):

$$S_{k,j} = \sum_{i=t_{0,k}}^{j} R_{k,i} \tag{1}$$

$$S_{k,j} \geqslant Y_k \tag{2}$$

Here, $S_{k,j}$ is the state on day $j$ of either aging, stress, or senescence ($k$) that are formulated as the sum of the corresponding rates on day $i$ ($R_{k,i}$), which accumulated between the starting day $t_{0,k}$ and $j$, until the threshold $Y_k$ is

190 reached. In other words, the daily aging rate ($R_{Aging,i}$) accumulates from LU ($t_{0,Aging}$ = LU). The transition from young leaf to mature leaf occurs when $S_{Aging,j}$ reaches $Y_{Aging,1}$. Thus, day $j$ becomes $t_{0,Stress}$ and the accumulation of the daily stress rate ($R_{Stress,i}$) starts, while $R_{Aging,i}$ continues to accumulate. While the transition from mature leaf to old leaf occurs when $S_{Aging,j}$ reaches $Y_{Aging,2}$, senescence is either induced with this transition or already earlier due to $S_{Stress,j}$ reaching $Y_{Stress}$. Upon senescence induction, day $j$ becomes $t_{0,Senescence}$ and the daily senescence rate ($R_{senescence,i}$) starts to accumulate. Eventually,

$S_{Senescence,j}$ reaches $Y_{LS50}$ and $Y_{LS100}$, and respective LS$_{50}$ and LS$_{100}$ are marked by the corresponding days $j$.

$R_{Aging,i}$ was either set equal to the daily net photosynthetic activity or to one (i.e., $A_{net}$ [mol C d$^{-1}$] or 1 [d d$^{-1}$], respectively), depending on the formulation (Eq. 3):

$$R_{Aging,i} = \begin{cases} A_{net,i} \\ 1 \end{cases} \tag{3}$$

$R_{Stress,i}$ was formulated as the sum of three to seven weighted stressors ($D_{stress}$; Eqs. 4–6), always considering (1) cold days (derived from minimum temperature; $Tn$ [°C]), (2) shortening days (derived from the difference in day length; $\delta L$ [h], with $\delta L_i = L_i - L_{i-1}$), and (3) dry days (approximated by the Keetch and Byram drought index [KBDI]; $Q$). In addition, some formulations of $R_{Stress}$ also considered (4) periods of heavy rainfall (approximated by the five-days

precipitation; $P5$ [mm], with $P5_i$ being the sum of $P_i$ to $P_{i-4}$), (5) heat days (derived from maximum temperature; $Tx$ [°C]), (6) nutrient depletion (approximated by the accumulated $A_{net}$ since LU, due to the absence of soil data), and/or (7) frost days (derived from lower minimum temperature than for cold days; Table S3; $Tn$ [°C]):

$$R_{Stress,i} = \sum w_{D_{Stress}} \times f(D_{Stress,i}) \tag{4}$$

$$D_{Stress,i} \in \left\{ Tn_i, \quad \delta L_i, \quad Q_i, \quad P5_i, \quad Tx_i, \quad \sum_{l=d_{LU}}^{i} A_{net,l}, \quad Tn_i \right\} \tag{5}$$

$$f(x) = \begin{cases} g(x) \\ h(x) \end{cases} \tag{6}$$

Here, $w_{D_{Stress}}$ is the weight for the response [$f(x)$] to $D_{Stress}$, calculated according to $g(x)$ or $h(x)$ (Eqs. 7 and 8):

$$g(x)=\begin{cases} 1 & , \quad if\ x \geq a \\ 0 & , \quad if\ x < a \end{cases}$$ (7)

$$h(x)=\begin{cases} 1 & , \quad if\ x < b_0 \\ \dfrac{b_1-x}{b_1-b_0} & , \quad if\ b_0 \leq x \leq b_1 \\ 0 & , \quad if\ x > b_1 \end{cases}$$ (8)

While $a$ marks the sudden boundary between an unstressed and stressed state, $b_0$ and $b_1$ mark the lower and upper bounds, respectively, between which stress gradually increases (Fig. 4). Because $x \geq a$ and $x \geq b_0$ result in stress, the response to $\delta L$ and $Tn$ was formulated as $g(-\delta L)$ and $g(-Tn)$ as well as $h(-\delta L)$ and $h(-Tn)$, which translates in stress if $\delta L \leq -a \vee -b_0$ and $Tn \leq -a \vee -b_0$. For example, if stress occurs suddenly or gradually when $\delta L \leq -0.01$ h, then

$a = 0.01$ h or $b_0 = 0.01$ h, respectively. Note that these are examples, see Table 3 for the calibrated values.

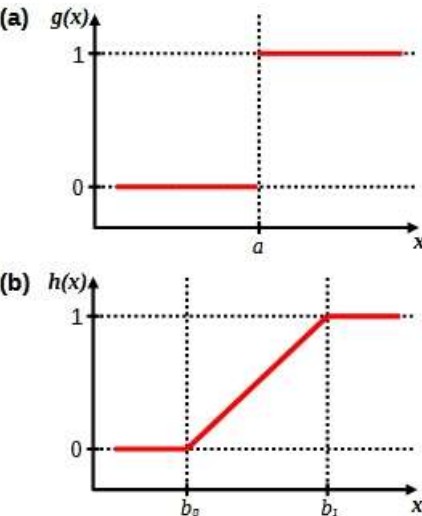

**Figure 4. Response functions (solid red lines) of $g(x)$ and $h(x)$.** In (a), $a$ marks the boundary value of $x$ at which $g(x)$ suddenly changes from 0 to 1 (i.e., from no effect to an effect). In (b), $b_0$ and $b_1$ mark the lower and upper bounds of $x$, respectively, between which $f(x)$ gradually increases from 0 to 1. Dotted lines are auxiliary lines.

$R_{Senescence,i}$ was either formulated as the sum, product, or exponential function of $R_{Aging,i}$ and $R_{Stress,i}$ or $S_{Aging,i}$ and $R_{Stress,i}$, which yield linear, convex, and sigmoid curves, respectively (Eq. 9):

$$R_{Senescence,i}=\begin{cases} w_A R_{Aging,i} + w_S R_{Stress,i} \\ s_X \left( R_{Aging,i} \times R_{Stress,i}^{x_S} \right) \\ s_X \dfrac{1}{e^{c\,S_{Aging,i}(d-R_{Stress,i})}} \end{cases}$$ (9)

$w_A$ and $w_S$ are the weights of $R_{Aging}$ and $R_{Stress}$, respectively, $s_X$ is a scaling factor, all of which allowed us to hard code $Y_{LS_{50}} = 1$, $x_S$ is the range bounded exponent of $R_{Stress}$, while $c$ and $d$ are the parameters of the sigmoid curve that relates $R_{Stress}$ and $S_{Aging}$ (Lang et al., 2019).

## 2.4        Model calibration and validation

We selected the observations for the calibration and validation samples with different procedures. To have a low risk of overfitting (i.e., the bias–variance trade-off; Sect. 2.2.2 in James et al., 2017), each calibration sample contained at least ten observations per calibrated parameter (Meier and Bigler, 2023). We defined two calibration datasets: one to calibrate a model that simulates both $LS_{50}$ and $LS_{100}$ simultaneously, and one to calibrate a model that simulates $LS_{50}$ only. For the two datasets, we selected site-years from those with the most extreme conditions during the growing season, i.e., the hottest, coldest, driest ten day periods observed between LU and $LS_{50}$ as well as the shortest and longest growing season observed in the pre-selected data (Sect. 2.1). For the first dataset, hereafter called '$LS_{50}$-$LS_{100}$ sample', we selected 250 of these site-years containing an observation for both $LS_{50}$ and $LS_{100}$. For the second dataset, hereafter referred to as '$LS_{50}$ sample', we selected 250 of these site-years containing observations for $LS_{50}$. These calibration samples were paired with validation samples that contained all remaining $LS_{50}$ and $LS_{100}$ observations or all remaining $LS_{50}$ observations, respectively. We drew twice both the $LS_{50}$ and $LS_{50}$-$LS_{100}$ samples. While model development was based on the $LS_{50}$-$LS_{100}$ samples, model evaluation was based on the $LS_{50}$ sample to allow for a comparison with previously published models. All models were calibrated five times per drawn sample (i.e., ten 'calibration runs' per model and $LS_{50}$ sample or $LS_{50}$-$LS_{100}$ sample) by minimizing the root mean squared error (RMSE; Eq. S44) with generalized simulated annealing and optimal, model-specific controls (see Sect. S2.2; Xiang et al., 1997, 2017).

## 2.5        Model development

We based our model on the most accurate formulation of the leaf development process after testing different formulations in several iterations (Fig. 5; see Table S3 for parameter ranges). First, we defined the process structure based on the stressors for cold days, shortening days, and dry days transformed through threshold response functions [$g(x)$; Eq. 7]. In iteration 1, we tested the definition of the aging rate ($R_{Aging}$) and of the senescence rate ($R_{Senescence}$). $R_{Aging}$ was formulated as a function of either the net photosynthetic activity ($A_{net}$) or of the number of days (Eq. 3). $R_{Senescence}$ was formulated as a combination of aging and stress [through $R_{Aging}$ or the state of aging ($S_{Aging}$) and through the stress rate ($R_{Stress}$)] in either a sum, product, or exponential function (Eq. 9). In iteration 2, we tested the number of phases of leaf development, i.e., either two phases 'mature leaf' and 'old leaf', or three phases 'young leaf', 'mature leaf', and 'old leaf'. Thus, we formulated $R_{stress}$ with a forward selection of additional stressors and selecting between $g(x)$ and gradual response functions [$h(x)$; Eq. 8]. In iteration 3, we considered each stressor for cold days, shortening days, and dry days through $h(x)$ rather than $g(x)$. In iteration 4, we considered one additional stressor, i.e. heavy rain periods, heat days, nutrient depletion, or frost days through $g(x)$. In iteration 5, we considered the additional stressor through $h(x)$ rather than $g(x)$. In iteration 6, the procedure of iterations 4 and 5 was repeated as long as they resulted in a formulation that was selected for further development.

       The formulations to be further developed were selected according to the accuracy of the corresponding model in simulating $LS_{50}$ and $LS_{100}$, i.e., through calibration with the $LS_{50}$-$LS_{100}$ sample. This accuracy was assessed with the Akaike information criterion corrected for small samples (AICc; Eq. S41; Akaike, 1974; Burnham and Anderson, 2004), which accounts for both the goodness-of-fit between the simulated and observed values and the number of free parameters. We calculated the AICc for each calibration run (see Sect. 2.4) and excluded the run with the highest AICc per model, before identifying the two models with the lowest median AICc across the given and all previous iterations. Finally, we selected the model based on the formulation with the lowest median AICc, which was further evaluated.

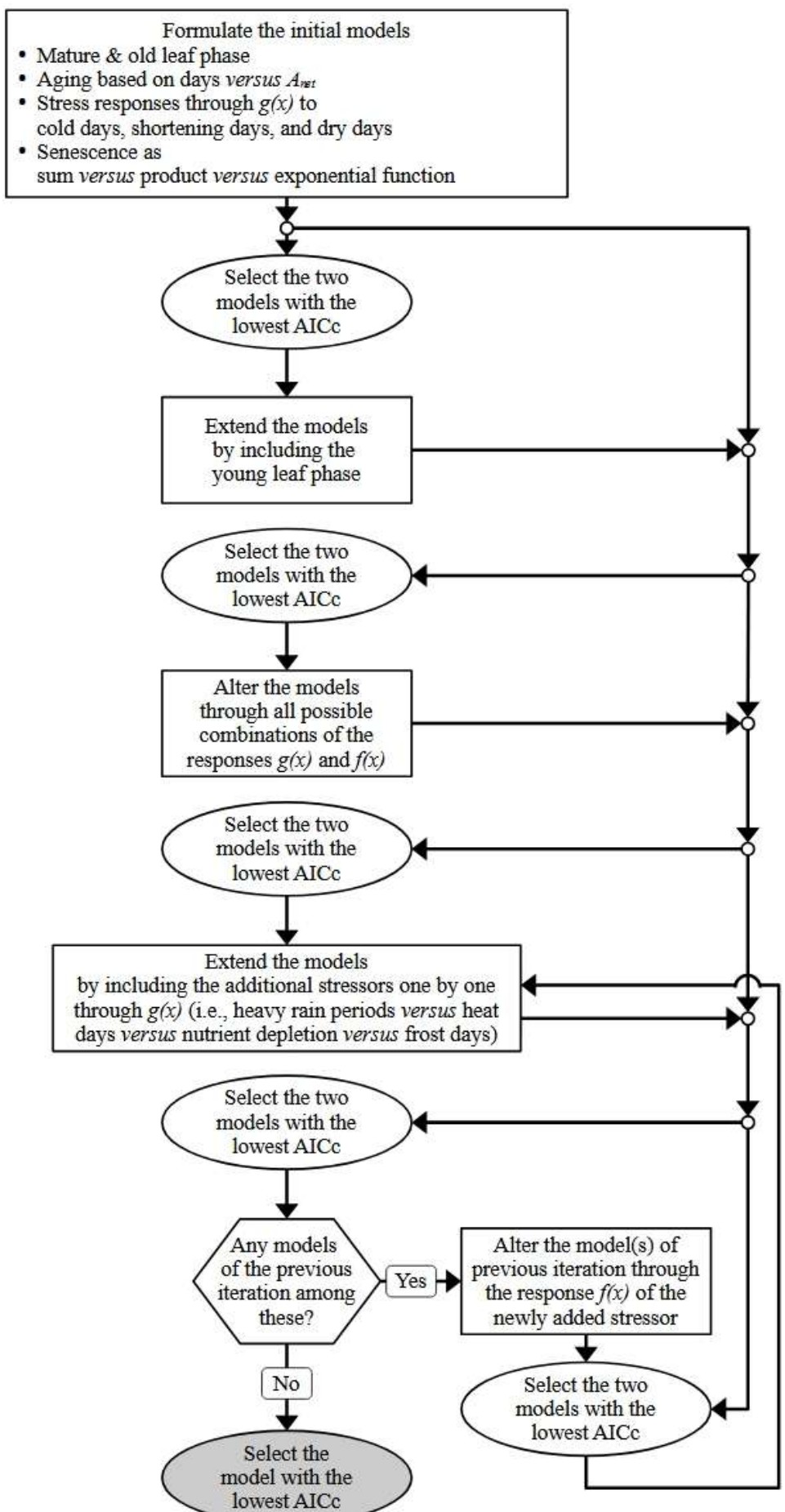

**Figure 5. Model development.** The iterations of model development are symbolized with rectangles. Selection of the best formulated models (ellipses) was based on the Akaike information criterion corrected for small samples (AICc; Eq. S41; Akaike, 1974; Burnham and Anderson, 2004) and the final selection is marked grey. For the response function to the stressors, i.e., *g(x)* and *h(x)*, see Eqs. 7 and 8.

## 2.6    Model evaluation

First, we evaluated the functionality of the selected model. We were particularly interested in the causes of senescence induction that could be due to aging or stress (Fig. 3). We counted how often aging versus stress induced senescence, and we quantified the relative amount of accumulated stress caused by each stressor at the time of senescence induction. We compared both aging- and stress-induced senescence as well as the relative amounts of stress across mean annual temperature (MAT; °C), mean annual KBDI (MAQ), latitude (LAT; °), and elevation (ELV; m a.s.l.) for the given year

and site. While MAT and MAQ are assumed to directly affect cold and dry stress, LAT relates to day length through the inclination angle of the Earth, and ELV relates to dry stress through decreasing nutrients with elevation (Huber et al., 2007; Loomis et al., 2006). The evaluation was based on the calibration runs that resulted in the highest modified Kling-Gupta efficiency (KGE'; Eq. S45; Gupta et al., 2009; Kling et al., 2012), which combines bias, variability, and correlation of the simulated and observed leaf senescence dates.

Second, we compared the accuracy between the selected model and three previously published models, namely the CDD, DM2, and PIA model. Because these models simulate only one stage of leaf senescence, which usually is $LS_{50}$, we based our comparison on this stage (Delpierre et al., 2009; Dufrêne et al., 2005; Zani et al., 2020). The CDD model determines $LS_{50}$ by the time the cold degree-days reaches a particular threshold (Dufrêne et al., 2005). The DM2 model accumulates the product of temperature differences and day length ratios to corresponding thresholds until

the threshold that determines $LS_{50}$ is reached (Delpierre et al., 2009). The PIA model accumulates temperatures and day lengths that are combined in an exponential function, and derives the threshold to determine $LS_{50}$ from the photosynthetic activity during the growing season (Zani et al., 2020). All these models were compared based on the calibration run that resulted in the highest KGE'. Further, we compared the RMSE and AICc as well as the Pearson correlation across the entire validation sample ($\rho_{Overall}$), across space ($\rho_{Spatial}$), and across time ($\rho_{Temporal}$). $\rho_{Spatial}$ was based on the site-

specific mean observed and simulated $LS_{50}$ across sites. $\rho_{Temporal}$ was calculated for each site, based on the yearly observed and simulated $LS_{50}$.

Third, we estimated the extent to which the model error (i.e., simulated minus observed $LS_{50}$) was affected by data structure as well as by climatic and spatial deviations from the $LS_{50}$ calibration sample, using a linear mixed-effects model (LMM; Pinheiro and Bates, 2000) and an analysis of variance (ANOVA; Sect. S2.4; Fox, 2016). In the

300 LMM, the response variable 'model error' was explained by the factor variable 'country' as well as the interaction of the factor variable 'model' with each of the differences between a site-year and the average of the calibration sample in MAT ($\delta$MAT), MAQ ($\delta$MAQ), the accumulated $A_{net}$ between LU and summer solstice ($\delta A_{net}$), latitude ($\delta$LAT), and elevation ($\delta$ELV). The random intercept was grouped by 'site'. The LMM was fitted with fast restricted maximum likelihood (Wood, 2011), and served as basis for an ANOVA. This type-III ANOVA (Yates, 1934) quantified the impact of

305 the explanatory variables in the variance of the model error that was explained by the LMM. The impact attributable to data structure was caused by the fixed effects of 'country' and the standard deviation in the random intercepts grouped

by 'site', while the impacts attributable to climatic versus spatial deviations from the calibration sample was caused by the effects of $\delta$MAT, $\delta$MAQ, and $\delta A_{net}$ versus the effects of $\delta$LAT and $\delta$ELV, respectively.

## 2.7 Statistical software and reporting of results

We used the programming language R, together with the R package data.table for data processing (Barrett et al., 2024). In R, data from xslx files were extracted with the R package readr (Wickham et al., 2024), and data from netCDF files were extracted and averaged with the R packages ncdf4 (Pierce, 2023), raster (Hijmans, 2023), sf (Pebesma, 2018; Pebesma and Bivand, 2023), and sp (Bivand et al., 2013; Pebesma and Bivand, 2005). Leap years were identified with the function leap_year in the R package lubridate (Grolemund and Wickham, 2011). Gaps in the regressed lapse rates

were filled with the function na.spline in zoo (Zeileis and Grothendieck, 2005). Seasonal splines of atmospheric $CO_2$ concentrations were calculated with the function sm in npreg (Helwig, 2024). The leaf senescence models were calibrated with the R package GenSA (Xiang et al., 2013), while the LMM was fitted with the R package mgcv (Wood, 2017) and the ANOVA was calculated with the R package stats (R Core Team, 2025). LMM estimates and 99% confidence intervals (i.e., significance level $a = 0.01$) for combined coefficients, e.g., the effect of $\delta$MAT for a given model,

were calculated with the Delta method (Fox and Weisberg, 2019, Chpt. 5.1.4; Wasserman, 2004, Chpt. 9.9) through the function deltaMethod in the R package car (Fox and Weisberg, 2019). For each LMM coefficient and ANOVA impact, we expressed the most optimistic change of odds between the null hypothesis (being zero; $H_0$) and alternative hypothesis (being different from zero or greater than zero, respectively; $H_1$) with the minimum Bayes factor ($\underline{BF}_{01}$), labeling $H_0:H_1$ ratios of 1/1000 and 1/100 as 'decisive' and 'very strong', respectively (Held and Ott, 2018; Johnson, 2005). $\underline{BF}_{01}$

was calculated from the $p$-values and number of data with the function tCalibrate in the R package pCalibrate (Held and Ott, 2018). For the visualizations, we used the R packages ggplot and ggpubr (Kassambara, 2020; Wickham, 2016), as well as the R packages ggspatial and rnaturalearth for the maps (Dunnington, 2023; Massicotte and South, 2023).

## 3 Results

### 3.1 Model formulation – the DP3 model

We tested 34 formulations of the leaf development process through 1428 calibration runs, and found that three subse-
330 quent leaf development phases resulted in the most accurate model (according to the AICc; Figs. 6 and S1–S2). In this model, the phase 'young leaf' starts with leaf unfolding. As a daily aging rate $R_{Aging}$ accumulates (Eq. 10), the simulated state of aging increases by one day per day. When this state reaches the threshold $Y_{Aging,1}$ (Eqs. 1 and 2), the phase 'mature leaf' begins. During this phase, the leaf continues to age and is also sensitive to stress caused by cold days, shortening days, and dry days, to which we hereafter refer to as 'cold stress', 'photoperiod stress', and 'dry stress', respectively.
This stress is summarized in a daily stress rate ($R_{Stress}$; Eq. 11) and thus accumulated to determine the state of stress. The first day that either the state of stress or the state of aging reaches the respective thresholds $Y_{Stress}$ or $Y_{Aging,2}$ (Eqs. 1 and 2), senescence is induced, while the phase 'old leaf' starts only when the state of aging reaches $Y_{Aging,2}$. Once senescence is induced, a daily senescence rate ($R_{Senescence}$) accumulates (Eq. 12) and determines the state of senescence. The days this state reaches the thresholds $Y_{LS50}$ and $Y_{LS100}$ (Eqs. 1 and 2) correspond to the simulated dates of $LS_{50}$ and $LS_{100}$, respec-
tively. Hereafter, we refer to this model as 'DP3' model (Tables 2 and 3; Meier, 2025b, coded in R).

$$R_{Aging,i} = 1 \tag{10}$$

$$R_{Stress,i} = w_C \, g(-Tn_i) + w_P \, g(-\delta L_i) + w_D \, g(Q_i) \tag{11}$$

$$R_{Senescence,i} = s_X \, R_{Stress,i}^{x_S} \tag{12}$$

Here, $w_C$, $w_P$, and $w_D$ are the weights for the response functions $g(x)$ (Eq. 7) to the minimum temperature ($Tn$), difference in day length ($\delta L$), and the Keetch and Byram drought index ($Q$) on day $i$, respectively [e.g., $w_P \, g(-\delta L_i)$ results in the photoperiod stress on day $i$]. $s_X$ is the scaling factor for $R_{Stress}$, which is 'shaped' by $x_S$.

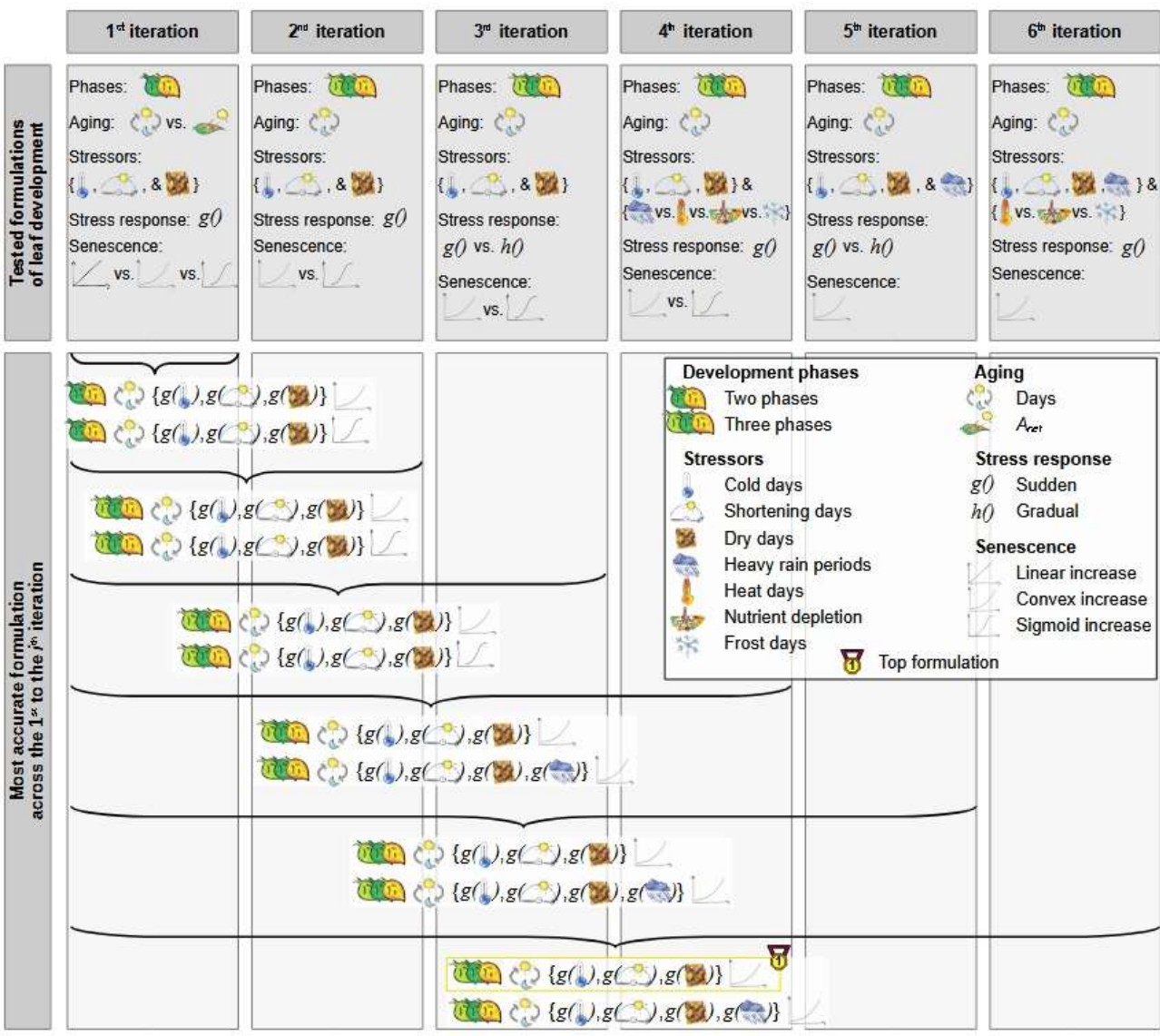

**Figure 6. Tested model formulations.** The tested formulations differed in their number of leaf development phases (i.e., two or three phases), in their driver of the aging rate (i.e., days or photosynthetic activity [$A_{net}$]), their stress rate in response (i.e., $g(x)$ or $h(x)$) to the stressors cold, shortening, dry, heat, and frost days, heavy rain periods, and nutrient depletion, and their response of the senescence rate to increasing age and stress (i.e., linear, convex, and sigmoid increase as a the result of a sum, product, or exponential function, respectively). After each iteration, we identified the two most accurate formulations across the given and all previous iterations (Fig. 5, Sect. 2.5). These formulation were further developed through the next iteration. As soon as an iteration did not produce any new model formulations, we selected the more accurately formulated model ('top formulation'; i.e., the 'DP3' model). All formulations were tested for beech based on the $LS_{50}$-$LS_{100}$ sample (Sect. 2.4).

**Table 2. Input and output variables of the DP3 model**

|  | Collective list | Name | Definition | Unit | Format |
|---|---|---|---|---|---|
| Input | - | par | Model parameters (see Table 3) | - | Vector |
|  | data | LU | Observed timing of leaf unfolding | doy | Vector |
|  |  | id | Unique identifier of each 'LU' (character) | - | Vector |
|  |  | $D_i$ | Daily number of days (i.e., 1 per day) | - | Matrix |
|  |  | $Tn_i$ | Daily minimum temperature | °C | Matrix |
|  |  | $\delta L_i$ | Daily difference in day length to previous day | h | Matrix |
|  |  | $Q_i$ | Daily Keetch and Byram drought index | - | Matrix |
|  | - | stages | Leaf senescence stages to be predicted (character, defaults to $LS_{Default}$) | - | Vector |
| Output | - | LS | Simulated leaf senescence dates, including senescence induction | doy | Matrix |
|  | transitions | $d_{y \to m}$ | Simulated timing of transition from young to mature leaf | doy | Vector |
|  |  | $d_{m \to o}$ | Simulated timing of transition from mature to old leaf | doy | Vector |
|  | aging | $R_{Aging,i}$ | Daily rate of aging | - | Matrix |
|  |  | $S_{Aging,i}$ | State of aging (i.e., accumulated $R_{Aging,i}$ since $LU$) | - | Matrix |
|  | stress | $X_{Cold,i}$ | Daily cold stress [i.e., $w_C\,g(-Tn_i)$] | - | Matrix |
|  |  | $X_{Photoperiod,i}$ | Daily photoperiod stress [i.e., $w_P\,g(-\delta L_i)$] | - | Matrix |
|  |  | $X_{Dry,i}$ | Daily dry stress [i.e., $w_D\,g(-Q_i)$] | - | Matrix |
|  |  | $R_{Stress,i}$ | Daily rate of stress | - | Matrix |
|  |  | $S_{Stress,i}$ | State of stress (i.e., accumulated $R_{Stress,i}$ since $d_{m \to o}$) | - | Matrix |
|  | senescence | $R_{Senescence,i}$ | Daily rate of senescence | - | Matrix |
|  |  | $S_{Senescence,i}$ | State of senescence (i.e., accumulated $R_{Senescence,i}$ since $d_{m \to o}$) | - | Matrix |

*Note*: Daily variables refer to day *i*, and accumulated variables refer to the period until day *i*. The vector par contains the model parameters listed in Table 3. In the collective lists data, aging, stress, and senescence, the rows of the matrices refer to the days of the year, while the columns refer to site-years and are ordered identically between all matrices. The order of these matrix columns matches the order of the vectors in the collective lists data and transition. For the LS matrix, the rows refer to the site-years and the columns refer to the senescence induction date (SI) and the dates of the leaf senescence stages indicated by the vector stages (Meier, 2025b).

**Table 3. Fitted parameters of the DP3 model**

| | | Fitted value | |
|---|---|---|---|
| Parameter | Meaning | $LS_{50}$-$LS_{100}$ | $LS_{50}$ |
| $-a_C$ | Boundary below which cold stress is 1 versus 0 (referring to $Tn_i$) | 2.55 °C | 0.06 °C |
| $-a_P$ | Boundary below which photoperiod stress is 1 versus 0 (referring to $\delta L_i$) | –0.0587 h | –0.0016 h |
| $a_D$ | Boundary above which dry stress is 1 versus 0 (referring to $Q_i$) | 176.94 | 183.82 |
| $w_C$ | Weight of cold stress | 0.14 | 0.29 |
| $w_P$ | Weight of photoperiod stress | 0.02 | 0.52 |
| $w_D$ | Weight of dry stress | 0.22 | 0.05 |
| $s_X$ | Scaling factor of the senescence rate | 0.59 | 0.35 |
| $x_S$ | Shape parameter of the stress rate | 0.21 | 5.67 |
| $Y_{Aging,1}$ | Age threshold for the transition from young to mature leaf | 41.59 d | 1.57 d |
| $Y_{Aging,2-Aging,1}$ | The threshold of aging during the mature leaf phase | 137.31 d | 71.58 d |
| $Y_{Aging,2}$ | Theoretical age threshold for the transition from mature to old leaf | 178.90 d | 73.14 d |
| $Y_{LS100}$ | Senescence threshold for $LS_{100}$ (all leaves have changed color or have fallen) | 5.95 | - |

*Note*: The parameters refer to the equations 7 and 9–11 and were fitted for beech with the $LS_{50}$ and $LS_{50}$-$LS_{100}$ sample (Sect. 2.4). All parameters were calibrated within the initial ranges (Table S3) to their fitted value. To avoid fitted values of $Y_{Aging,1} > Y_{Aging,2}$, we used and calibrated $Y_{Aging,2-Aging,1}$ instead of $Y_{Aging,2}$. The theoretical threshold $Y_{Aging,2}$ was not calibrated but calculated from $Y_{Aging,1} + Y_{Aging,2-Aging,1}$ and displayed for easier interpretation. The thresholds for stress ($Y_{Stress}$) and $LS_{50}$ ($Y_{LS50}$; i.e., the time when 50% of the leaves have changed color or have fallen) were hard coded with $Y_{Stress} = 1$ and $Y_{LS50} = 1$. The shortening of day length of 0.0016 h ($a_P$; i.e., 0.1 minutes) based on the $LS_{50}$ calibration is breached on doy 175, 174, and 174 (i.e., June 24, 23, and 23) at the exemplary minimum, median, and maximum latitudes of our samples (i.e., 45.9°, 47.8°, and 58.0° north), respectively. Alternatively, the shortening of 0.0587 h (3.5 minutes) based on the $LS_{50}$-$LS_{100}$ calibration is breached on doy 252 and 202 (i.e., September 9 and July 21) at the exemplary median and maximum latitudes of our samples, respectively, whereas it is never breached at the exemplary minimum latitude.

According to the DP3 model, leaf senescence was generally induced earlier during warmer years and at lower elevations (Fig. 7; Tables S5–S8). In average, senescence was induced a month earlier when mean annual temperatures were 13–15 °C than when they were 4–6 °C (i.e., May 29 versus June 22 and April 20 versus May 12 when the DP3 model was calibrated with the $LS_{50}$-$LS_{100}$ and $LS_{50}$ samples, respectively; hereafter referred to as 'DP3$_{LS_{50}\text{-}LS_{100}}$ model' and 'DP3$_{LS_{50}}$ model'; Sect. 2.4). Accordingly, senescence induction was 20 days earlier below 288 m a.s.l. than above 1150 m a.s.l. (i.e., June 5 versus June 25 and April 26 versus May 16 based on the DP3$_{LS_{50}\text{-}LS_{100}}$ and DP3$_{LS_{50}}$ model, respectively). Both the DP3$_{LS_{50}\text{-}LS_{100}}$ model and DP3$_{LS_{50}}$ model predicted generally longer senescence (i.e., the duration between $LS_{50}$ or $LS_{100}$ and senescence induction) during years of higher mean annual temperatures (Fig. S3; Tables S5–S8).

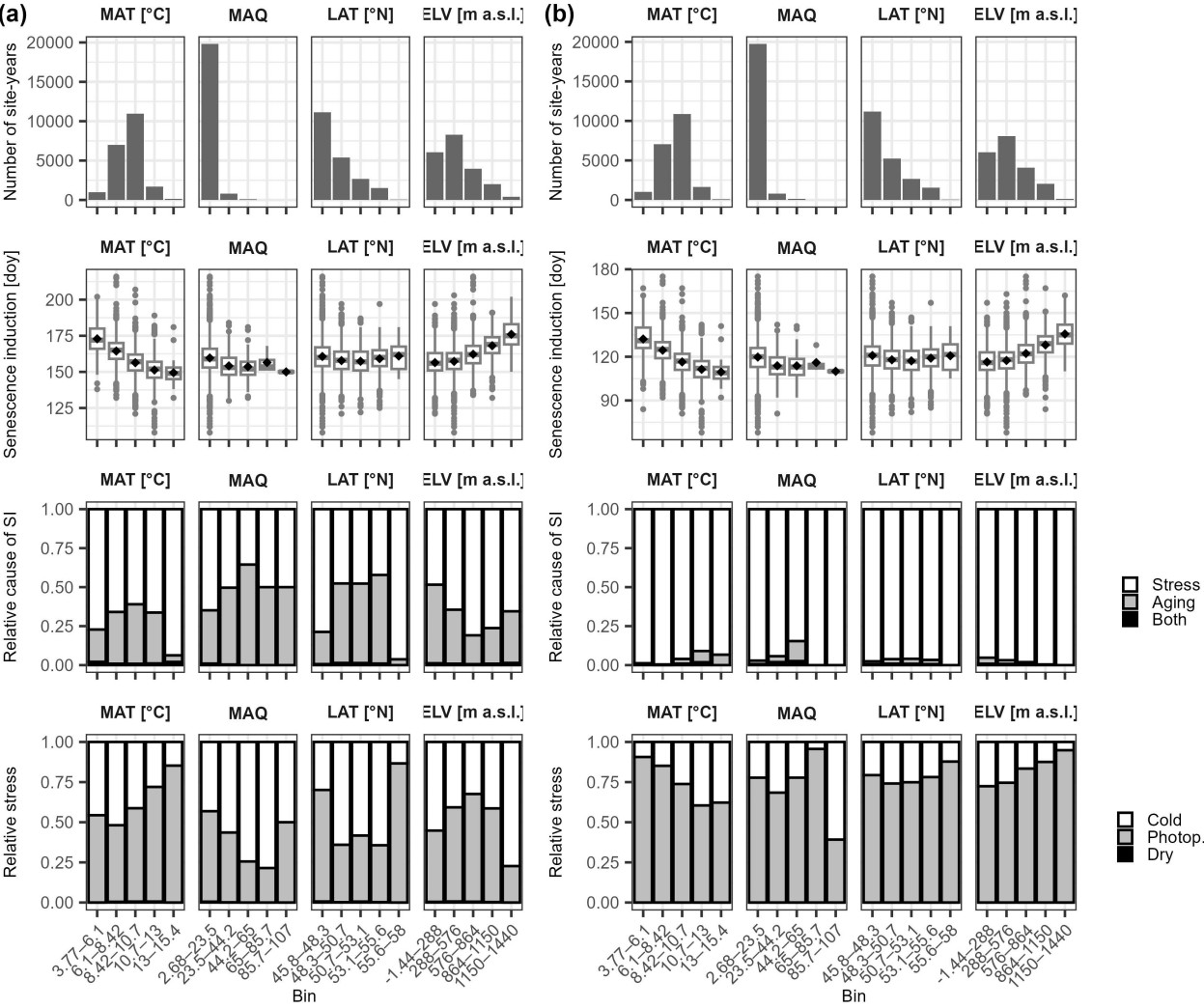

**Figure 7. Senescence induction.** Panel (a) and (b) are based on simulations by the DP3 model calibrated with the $LS_{50}$ versus $LS_{50}$-$LS_{100}$ sample, respectively (Sect. 2.4). The top row of each panel shows the number of site-years in the bins, which were equally distributed among mean annual temperature (MAT, °C), mean annual Keetch and Byram drought index (MAQ), latitude (LAT, °N), and elevation (ELV; m a.s.l.). The second row of each panel visualizes the date of senescence induction in day of year [doy]. While the mean and median dates are marked with black dots and grey lines, respectively, the most extreme values are indicated with dots if outside ±1.5 times the inner quartile range from the 1st and 3rd quartile, and with whiskers otherwise. The third row of each panel illustrates the relative number of site-years during which senescence was induced by stress versus aging or by both stress and aging [both the accumulated stress rate and accumulated aging rate reached their thresholds for senescence induction (SI) on the same date]. The bottom row of each panel shows the relative amount of cold, photoperiod (Photop.), and dry stress that accumulated at the time of senescence induction by stress.

Stress induced senescence two times and 40 times more often than aging according to the $DP3_{LS_{50}-LS_{100}}$ and $DP3_{LS_{50}}$ model, respectively (Fig. 7, Tables S5–S8). Thus, while aging was of negligible importance to senescence induction according to the $DP3_{LS_{50}}$ model, it mattered according to the $DP3_{LS_{50}-LS_{100}}$ model, particularly during years of medium mean annual temperature (6–13° C) as well as at sites of medium latitude (48.3–55.6 °N) and of low elevation (below 576 m a.s.l.). At the time of senescence induction due to stress, the amounts of accumulated photoperiod stress and cold stress relative to total stress were 56% versus 44% ($DP3_{LS_{50}-LS_{100}}$ model) and 77% versus 23% ($DP3_{LS_{50}}$ model), respectively, while the corresponding amounts of dry stress were 0.5% and 0.0%. Photoperiod stress dominated mostly in warm years and medium-elevation sites according to the $DP3_{LS_{50}-LS_{100}}$ model, whereas it did so in cool years and high-elevation sites according to the $DP3_{LS_{50}}$ model. In summary, photoperiod stress rather than cold and dry stress induced leaf senescence, but the importance of these stressors and their dependency on climatic conditions and location differed between the $DP3_{LS_{50}-LS_{100}}$ and $DP3_{LS_{50}}$ model.

Accordingly, the relative importance of these stressors for the duration of senescence differed between the $DP3_{LS_{50}-LS_{100}}$ and $DP3_{LS_{50}}$ model (Fig. S3; Tables S5–S8). Photoperiod stress clearly dominated the progress from senescence induction to $LS_{50}$ according to the $DP3_{LS_{50}}$ model. However, according to the $DP3_{LS_{50}-LS_{100}}$ model and especially during cool years, cold stress was most important between senescence induction and $LS_{50}$, whereas photoperiod stress was most important between senescence induction and $LS_{100}$.

## 3.2 Model accuracy

The DP3 model simulates leaf senescence with similar accuracy as previous models (Fig. 8; Table 4). All models calibrated with the $LS_{50}$ sample resulted in an RMSE of ~15 d, with the lowest RMSE for the Null model (i.e., constant prediction of the average observation in the calibration sample). The $LS_{50}$-$LS_{100}$ sample yielded considerable higher RMSE for both the $DP3_{LS_{50}-LS_{100}}$ and Null model, namely 23–25 d and 18–21 d, respectively. Nevertheless, the $DP3_{LS_{50}-LS_{100}}$ model resulted in the highest overall correlation ($\rho_{Overall}$ of 0.2 for $LS_{100}$). The highest correlation across space was obtained with the PIA model ($\rho_{Spatial}$ of 0.4), while the DP3 model resulted in the highest correlation across time (average $\rho_{Temproal}$ of 0.05 according to $DP3_{LS_{50}}$ and according to $DP3_{LS_{50}-LS_{100}}$ for $LS_{100}$).

**Table 4. Model accuracy**

| Model | Sample | Stage | KGE' | RMSE | AICc | $\rho_{Overall}$ | $\rho_{Spatial}$ | $\overline{\rho}_{Temporal}$ | n |
|---|---|---|---|---|---|---|---|---|---|
| CDD | $LS_{50}$ | $LS_{50}$ | -0.13 | 16.1 | 57797 | 0.01 | -0.09 | 0.04 | 6887 |
| DM2 | $LS_{50}$ | $LS_{50}$ | -0.26 | 15.0 | 56862 | 0.02 | -0.12 | 0.00 | 6887 |
| PIA | $LS_{50}$ | $LS_{50}$ | -0.19 | 14.8 | 56701 | 0.10 | 0.44 | -0.04 | 6887 |
| $DP3_{LS_{50}}$ | $LS_{50}$ | $LS_{50}$ | -0.23 | 15.2 | 57083 | 0.02 | -0.02 | 0.05 | 6887 |
| Null | $LS_{50}$ | $LS_{50}$ | NA | 14.8 | NA | NA | NA | NA | 6887 |
| $DP3_{LS_{50}-LS_{100}}$ | $LS_{50}$-$LS_{100}$ | $LS_{50}$ | -0.01 | 25.0 | 63911 | 0.04 | -0.06 | 0.03 | 6887 |
| $DP3_{LS_{50}-LS_{100}}$ | $LS_{50}$-$LS_{100}$ | $LS_{100}$ | 0.14 | 23.2 | NA | 0.22 | 0.17 | 0.05 | 600 |
| Null | $LS_{50}$-$LS_{100}$ | $LS_{50}$ | NA | 18.1 | NA | NA | NA | NA | 6887 |
| Null | $LS_{50}$-$LS_{100}$ | $LS_{100}$ | NA | 21.7 | NA | NA | NA | NA | 600 |

*Note*: The Null model constantly predicts the average observation in the calibration sample (i.e., either the stage when 50% or 100% of the leaves have changed color or have fallen; $LS_{50}$ or $LS_{100}$, respectively). The modified Kling-Gupta efficiency (KGE'), root mean squared error (RMSE), Akaike information criterion for small samples (AICc), and Pearson correlation overall, across space, and across time [$\rho_{Overall}$, $\rho_{Spatial}$, and average $\rho_{Temporal}$ ($\overline{\rho}_{Temporal}$), respectively] are explained in Sect. 2.6, S2.1, and S2.2. All these metrics were calculated for the simulations and observations of the validation samples $LS_{50}$ and $LS_{50}$-$LS_{100}$ (Sect. 2.4). Except the RMSE, they result in NA if the variance of the simulated values is zero, which is the case for the Null model. n indicates the number of observations in the validation sample. In addition, the AICc for the stage $LS_{100}$ according to the model $DP3_{LS_{50}-LS_{100}}$ was omitted because n differed between $LS_{100}$ and $LS_{50}$.

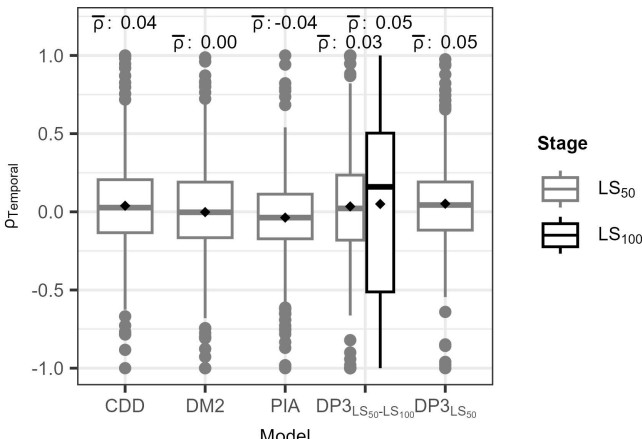

**Figure 8. Temporal Pearson correlation ($\rho_{Temporal}$).** The distribution ot the Pearson correlation within site ($\rho_{Temporal}$) between simulated and observed leaf senescence [the dates when 50% and 100% of the leaves have changed color or have fallen ($LS_{50}$ and $LS_{100}$, respectively)] is displayed for each model. The DP3 model was calibrated twice, namely with the $LS_{50}$-$LS_{100}$ sample ($DP3_{LS_{50}-LS_{100}}$) and with the $LS_{50}$ sample ($DP3_{LS_{50}}$; Sect. 2.4), the latter of which was also used to calibrate the CDD, DM2, and PIA model. The mean $\rho_{Temporal}$ (black rhombuses) is indicated above each box ($\bar{\rho}$). The boxes indicate the inner quartile range and the median (middle line). The most extreme values are indicated with dots if outside ±1.5 times the inner quartile range from the 1st and 3rd quartile, and with whiskers otherwise.

### 3.3 Model error

The model errors according to the DP3 model and previous models were similarly affected by data structure and climatic and spatial deviations from the calibration sample as the Null model (Fig. 9). The data structure was described by the fixed effects of countries and the random intercepts grouped by sites. The countries altered the model error by –18 to +8 d, depending on the model (Tables S9–S10). The standard deviation in the model error due to the random intercepts was 9 d. Depending on the model, the fixed effects of the climatic deviations ranged from –22 to –19 d 10°C$^{-1}$ ($\delta$MAT), from +3.6 to +9.0 d 100$^{-1}$ ($\delta$MAQ), and from +4.1 to +4.6 d 10mol C$^{-1}$ ($\delta A_{net}$), respectively. The model-specific effects of the spatial deviations $\delta$LAT and $\delta$ELV ranged from +2.0 to +2.1 d °N$^{-1}$ and from +1.0 to +1.1 d 100m$^{-1}$, respectively. While the evidence in the data was decisive ([BF]$_{01}$ < 1/1000; Sect. 2.7) for an effect of the CDD model on the model error different from zero as well as for the individual climatic deviations and for $\delta$LAT. The evidence was significant ($p < 0.005$) for corresponding effects of the CDD and DM2 model as well as for all individual climatic and spatial deviations and for all individual countries. The evidence was neither decisive nor significant for any effect different from zero of the interaction terms between the models and the climatic or spatial deviations. The LMM explained the model error with an adjusted R$^2$ of 0.44. Differences among sites attributed for 92% of the variance in the model error explained by the LMM, followed by the effects of $\delta A_{net}$ and $\delta$MAT (6% and 2%, respectively), whereas the effects of the models accounted for 0.3% (Table S11). In general, the model errors according to the DP3 model and previous models behaved as those of the Null model and mainly varied due to data structure.

Earlier start of leaf senescence in warmer years

**(a)**

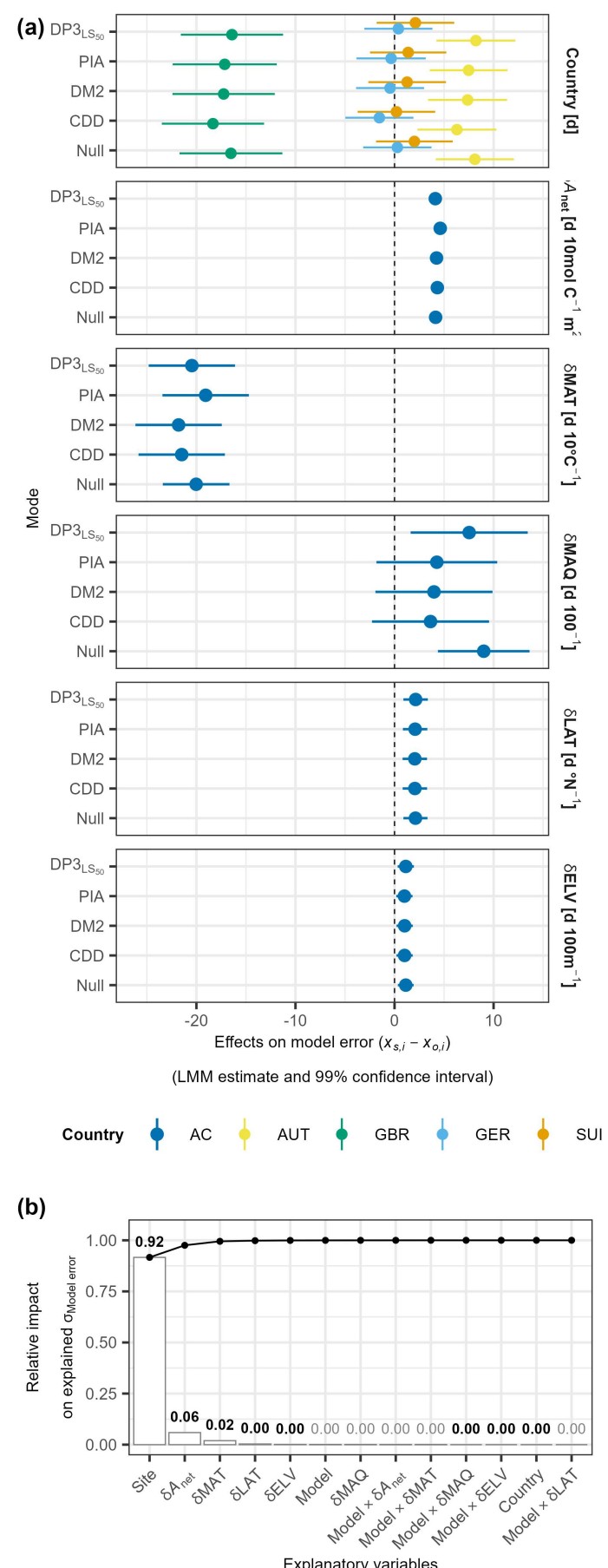

(LMM estimate and 99% confidence interval)

Country    AC    AUT    GBR    GER    SUI

**(b)**

**Figure 9. Model error versus data structure and climatic and spatial deviations.** Panel (a) visualizes the LMM-based, model-specific estimated fixed effects (dots) and 99% confidence intervals (bars) of data structure described by 'country', climatic deviations described by mean annual temperature ($\delta$MAT; d 10°C$^{-1}$), mean annual Keetch and Byram drought index ($\delta$MAQ; d 100$^{-1}$), and accumulated net photosynthetic activity between leaf unfolding and summer solstice ($\delta A_{net}$; d 10mol C$^{-1}$), and spatial deviations described by latitude ($\delta$LAT; d °$^{-1}$) and elevation ($\delta$ELV; d 100m$^{-1}$). These deviations were calculated as the difference between a given site-year and the average in the calibration sample. The colors indicate the countries Austria (AUT), Great Britain (GBR), Germany (GER), and Switzerland (SUI) as well as estimates across countries (AC). The model error was calculated as the simulated minus the observed timing ($x_{s,i} - x_{o,i}$). Panel (b) shows the relative impact of the explanatory variables on the variance in the model error as explained by the LMM. The random intercepts in the LMM were grouped by 'site', also describing data structure. The bars indicate the impact of individual variables, while the connected dots show the accumulated impact. The numbers above each bar state the impact, being bold in case of combined significance and decisiveness (i.e., $p \leq 0.01$ and minimum Bayes factor $\leq 1/1000$).

## 4          Discussion

### 4.1          Model formulation

The DP3 model simulates leaf senescence dates through a novel formulation that differs considerably from the formulation of current models. This novel formulation may change the way we see leaf senescence, namely as a consequence of leaf development that relates to both aging and stress. Current models start their simulation on the senescence induction date, which they determine from day length and temperature (e.g., Delpierre et al., 2009; Dufrêne et al., 2005; Keenan and Richardson, 2015; Lang et al., 2019; Liu et al., 2019; Zani et al., 2020). This date is calibrated such that leaf senescence dates are simulated most accurately. In the DP3 model and in addition to this prerequisite, at least accumulated aging or accumulated stress since leaf unfolding must have reached a given threshold. In other words, while current models define the senescence induction date backward, the DP3 model defines it both backward and forward, arguably resulting in a more robust definition. Moreover, as current models generally ignore aging (but see the model by Keenan and Richardson, 2015, which considers the leaf unfolding date in the stress threshold for leaf senescence), their formulation partially ignores current knowledge (e.g., Field and Mooney, 1983; Guo et al., 2021; Jibran et al., 2013; Lim et al., 2007). In addition, the models by Liu et al. (2019) and Zani et al. (2020) postulate an effect of the conditions before senescence induction on senescence duration, which remains speculative. However, the conditions before senescence induction likely affect senescence induction dates, possibly through photosynthetic activity (Zohner et al., 2023) or through aging and stress (DP3 model).

The novel formulation of the DP3 model supports the advancement of leaf senescence research by postulating new hypotheses. To our knowledge, it is the first process-based leaf senescence model that (a) simulates leaf senescence dates through daily leaf development status, (b) starts the simulation with leaf unfolding, (c) differentiates between daily aging and stress rates, and (d) predicts the dates of transition between the leaf developmental phases young, mature, and old leaf as well as the date of senescence induction. This allows the development of several new hypotheses (Carley, 1999; Hauke et al., 2020), which may relate to the currently disputed effect of climate change on productivity (Lu and Keenan, 2022; Norby, 2021; Zani et al., 2020; Zohner et al., 2023) and can be tested by controlled experiments. In particular, these hypotheses may concern (1) the duration of the young leaf phase during which stress cannot induce senescence, (2) the timing and cause (i.e., aging versus stress) of senescence induction, and (3) the relative importance of the stressors in relationship to climate and location, all of which are further elaborated here below

The duration of the young leaf phase differed considerably between the DP3$_{LS50-LS100}$ and DP3$_{LS50}$ model (i.e., the DP3 models calibrated with the LS$_{50}$-LS$_{100}$ versus LS$_{50}$ samples; Sect. 2.4), namely 41 d versus 1 d, respectively. Be-

cause the DP3 assumes that stress during this phase is irrelevant for senescence induction, the duration of this phase affects the date of the induction and end of leaf senescence (see below). Moreover, corresponding projections under future climate scenarios are also likely affected, as the probability of late spring frost events likely will change under climate warming (Bigler and Bugmann, 2018; Meier et al., 2018; Sangüesa-Barreda et al., 2021). Therefore, duration and characteristic of this young leaf phase should be examined further, e.g., with controlled experiments that apply continuous stress right after leaf unfolding to determine until when stress is either completely irrelevant for senescence induction or accumulates without inducing senescence.

Senescence was induced in late spring/early summer and more often by stress than by aging, but the induction dates and the stress:aging ratios differed notably between the $DP3_{LS_{50}-LS_{100}}$ and $DP3_{LS_{50}}$ model. The induction dates predicted by the $DP3_{LS_{50}-LS_{100}}$ and $DP3_{LS_{50}}$ models differed by 40 d, which matches the difference in the predicted duration of the young leaf phase (see above). As stress during the young leaf phase does not affect the predicted timing of leaf senescence by definition (Figs. 1, 3a, and 3c), this result illustrates the importance of studying the effects of stress after leaf unfolding. It also shows that different combinations of calibrated model parameters eventually yield similar predictions. Such compensating effects between different model parameters have also been reported in previous studies (Chuine and Régnière, 2017; Van der Meersch and Chuine, 2025), and explain the different stress:aging ratios as well as the earlier senescence induction during warmer years and at lower elevations. In warmer years and at lower sites, cold stress arguably decreases more (see below) than dry stress increases, while photoperiod stress remains unaffected, which decelerates senescence. On the one hand, senescence must be induced earlier in warmer years and at lower sites to predict leaf senescence dates that are biased to the mean and constant, as suggested by model accuracy and model error (see below). On the other hand, earlier induction and longer duration of senescence in warmer years may also be a valid description of reality (Yu et al., n.d.; Zohner et al., 2023). However, Zohner et al. (2023) argued that senescence induction dates relate negatively to pre-solstice productivity (see also Zani et al., 2020), whereas we showed that these dates relate to particular interactions between aging and stress rather than to productivity (see below; Eqs. 3 & 10; Lu and Keenan, 2022; Marqués et al., 2023; Norby, 2021). Because such different mechanisms very likely affect leaf senescence projections under climate warming, they certainly need further investigations.

How do aging and stress interact to predict earlier induction and longer duration of senescence in warmer years and at lower sites? The aging requirement for the transition from mature to old leaf (i.e., $Y_{Aging,2}$; Table 3) represents the longest possible duration from leaf unfolding to senescence induction. Earlier senescence induction is only possible through stress, which further relates negatively to the duration of senescence. At the same time, leaves unfold earlier at lower sites in general (Vitasse et al., 2009, 2013) and in warm springs in particular (given that the buds have been sufficiently chilled; Asse et al., 2018; Meier et al., 2021; Menzel et al., 2020). Warmer years have been shown to increase cold stress in spring (i.e., through leaves unfolding overly early in comparison to late frost; Asse et al., 2018; Meier et al., 2018; Sangüesa-Barreda et al., 2021) and relate positively to dry stress (i.e., through evapotranspiration; Allen et al., 1994; Berdanier and Clark, 2018; Wu et al., 2022), while leaving photoperiod stress unaffected (Brock, 1981). Thus, earlier senescence induction results from earlier leaf unfolding in combination with increased cold and dry stress during the mature leaf phase, while longer senescence duration relates to decreased stress during senescence.

Surprisingly at first, the $DP3_{LS_{50}-LS_{100}}$ model postulated photoperiod rather than cold and dry stress of being the most important stressor for senescence induction during warmer years, whereas the $DP3_{LS_{50}}$ model saw photoperiod stress of being most important during cooler years. By definition, stress only accumulates during the mature leaf phase but not during the young leaf phase (Figs. 1, 3a, and 3c). The threshold for photoperiod stress is likely reached only dur-

ing the mature leaf phase (i.e., after July 21 to September 9 in the $DP3_{LS_{50}-LS_{100}}$ model and after June 23–24 in the $DP3_{LS_{50}}$ model, depending on latitude; Table 3), and unless senescence is induced soon after this day by either cold or dry stress, photoperiod stress gains in importance quickly. The threshold for cold stress is likely reached in spring and autumn, and thus during both the young and mature leaf phases. Thus, the longer the young leaf phase and the later it ends, the less

likely late cold days in spring affect the senescence induction date and vice versa. In addition, the later aging may induce senescence at the end of the mature leaf phase and the later photoperiod stress starts to accumulate, the more cold days in fall can be accumulated and vice versa. Therefore, on the one hand, the long young leaf phase and late accumulation of photoperiod stress favor the accumulation of cold stress in autumn, which likely decreases in warmer years, making photoperiod stress relatively more important in the $DP3_{LS_{50}-LS_{100}}$ model. On the other hand, the short young leaf

phase favors the accumulation of cold stress in spring, which likely decreases in cooler years through leaves unfolding overly late in comparison to late frost (Asse et al., 2018; Meier et al., 2018; Sangüesa-Barreda et al., 2021), making photoperiod stress relatively more important in the $DP3_{LS_{50}}$ model.

### 4.2         Model accuracy

We compared the DP3 model to three previous models of leaf senescence (i.e., the models CDD, DM2, and PIA;

Delpierre et al., 2009; Dufrêne et al., 2005; Zani et al., 2020) based on the $LS_{50}$ calibration sample and found the RMSE of all compared models to be above the RMSE for the Null model (i.e. the constant prediction of the average observation in the calibration sample). This may be explained by unrealistic model formulations, poor model calibrations, and noisy data to drive and calibrate the models, all of which we discuss here below.

While the formulations of the compared models differ, they all build on the results of previous studies. For

example, according to all compared models, the leaf senescence date advances due to cold temperatures, which was also observed by Kloos et al. (2024), Wang et al. (2022), Wang and Liu (2023), and Xie et al. (2015, 2018). Moreover, in all but one model, shorter days cause earlier leaf senescence, which is is in agreement with Addicott (1968), Keskitalo et al. (2005), Singh et al. (2017), Tan et al. (2023), and Wang et al. (2022). Therefore, while the Null model predicted the leaf senescence dates more accurately according to the RMSE, it is unlikely that it is more realistically formulated than the

compared models. The currently most realistic model is arguably the DP3 model (Jan et al., 2019; Jibran et al., 2013; Lim et al., 2007), which makes it the first choice to study the leaf senescence process (see above). Moreover, while the Null model could be a good choice for predictions of leaf senescence dates (i.e., accuracy), the most suited models for predictions of leaf senescence trends (i.e., precision) may have to be identified yet.

We calibrated the compared models with the generalized simulated annealing algorithm and with model-spe-

500 cific controls (Sect. 2.4 and S2.1; Xiang et al., 1997, 2017). Algorithm and controls affect the accuracy of the calibrated models (Meier and Bigler, 2023). Therefore, we used generalized simulated annealing, which is a well established optimization algorithm and was shown to yield accurate models of leaf phenology (Chuine et al., 1998; Meier and Bigler, 2023) and has been used by many studies to calibrate such models (e.g., Basler, 2016; Liu et al., 2019; Meier et al., 2018; Zani et al., 2020). In addition, we used model-specific controls selected to most accurately simulate leaf senes-

505 cence dates for the validation samples (Sect. S2.2). Possible overfitting (James et al., 2017) through this procedure is unlikely, as the number of observations in the calibration samples was large enough (Sect. 2.4; Jenkins and Quintana-Ascencio, 2020; Meier and Bigler, 2023). Moreover, the compared models would have benefited from overfitting, as the comparison to the Null model was based on the same validation samples as the selection of the controls. Therefore, it is

510 highly improbable that this procedure caused the models to be calibrated so poorly that they are outperformed by the Null model.

All compared models were driven with daily weather data from the E-OBS dataset (Cornes et al., 2018) and calibrated and validated with leaf senescence data from the datasets of Meteo Swiss and PEP725 (Swiss phenology network, 2025; Templ et al., 2018). The E-OBS dataset has been used by many studies (e.g., Bowling et al., 2024; Meng et al., 2021; Schwaab et al., 2021; Zeng and Wolkovich, 2024), and we are unaware of any difficulties concerning the

515 daily weather data used here. The Meteo Swiss and PEP725 datasets, however, compile visually observed leaf senescence data, and such data is noisy due to different observers and small sample sizes (Liu et al., 2021): estimates of the leaf senescence dates for individual trees varied by 15 d (median, spreading from 2–53 d) between observers, and increased to 28 d (median) for different samples of ten trees. The data become even noisier if the observers follow different protocols from various institutions and countries (Menzel, 2013), eventually blurring the signal of the leaf senes-

520 cence process. Arguably the more this signal is blurred, the closer the simulations will follow the mean observation in the data. Here, we used leaf senescence data from 244 sites (i.e., at least 244 observers) and four countries (Sect. 2.1), which implies considerable noise and thus a blurred signal of the leaf development process. This data very likely forced the compared models to simulate leaf senescence dates close to the mean observation, impairing their accuracy.

### 4.3 Model error

While the model error was generally affected by climatic and spatial deviations from the calibration sample, their model-specific effects only differed insignificantly from the Null model. In other words, the model error in the compared models reacted similarly to climatic and spatial deviations as the model error of the Null model. This implies that the compared models simulated leaf senescence dates closely to the mean observation of the calibration sample and thus were heavily biased to the mean (i.e., as the Null model). Possible explanations for this are unrealistic model formula-

tions, poor model calibrations, and noisy data. Interestingly, Meier et al. (2023), who reported a heavy bias towards the mean for 21 process-oriented models of leaf senescence, based their study on leaf senescence data from 500 sites (i.e., at least 500 observers) and at least three countries from the PEP725 dataset (Templ et al., 2018). This supports our inference that the compared models resorted to the mean observation due to the used leaf senescence data rather than to model formulations and model calibrations.

Leaf senescence data was most relevant for the model error in the compared models, which was illustrated by the fixed effects of countries and the variation caused by the random intercepts grouped by sites. These effects of countries have, to our knowledge, not been studied yet, and differed considerably between countries, which demonstrates the noise added to leaf senescence data by different observation protocols (see above; Menzel, 2013). The random intercepts grouped by sites varied considerably, and corresponding differences among sites were attributed to a substantial amount of the explained variance in the model error (Chpt. 23.3.2 in Fox, 2016). Meier et al. (2023) also noted a large

amount of the explained variance in the RMSE being attributed to differences among sites. They reasoned that this was caused by, among others, noisy leaf senescence data (see above) and different inter-annual variability of observations between the sites (Cole and Sheldon, 2017; Čufar et al., 2015; Li et al., 2022; Liu et al., 2020). It remains to be seen if such site-specific inter-annual variability as well as inter-site variability in leaf senescence dates would be simulated

correctly by models calibrated with noise-free data.

## 4.4     Ways forward

While the DP3 model is likely the currently most realistic process-oriented model of leaf senescence, it may be developed further by (1) testing other drought indices, (2) considering nutrient depletion in combination with drought, and (3) ameliorating the formulation of the senescence rate. First, while various indices summarize drought differently (Speich, 2019; Zargar et al., 2011), the KBDI used here can be calculated from few data, being based on precipitation and temperature. It should be tested, however, if other indices, such as the standardized precipitation evapotranspiration index (based on precipitation and temperature; Vicente-Serrano et al., 2010) or the ratio of actual to potential evapotranspiration (based on precipitation, temperature, and soil moisture; Bugmann and Cramer, 1998), may approximate the effects of dry stress on leaf senescence more accurately. Second, despite more accurate simulations of $LS_{50}$ and $LS_{100}$ when nutrient depletion was disregarded (Figs. 6 and S1), model errors indicated earlier observed than simulated $LS_{50}$ and $LS_{100}$ dates due to nutrient depletion as approximated by elevation (Fig. 9; Tables S9–S10). This can be explained by higher elevation relating to increased nutrient depletion, which in turn fuels dry stress (Fu et al., 2014; Huber et al., 2007; Loomis et al., 2006; Tan et al., 2023). Consequently, drought indices that consider nutrient depletion should be tested. Third, the $DP3_{LS_{50}\text{-}LS_{100}}$ model was considerably less accurate than the $DP3_{LS_{50}}$ model, implying difficulties in the accurate and simultaneous simulation of $LS_{50}$ and $LS_{100}$. This points to an incorrectly formulated curve of the senescence rate (Eqs. 1 & 12), and corresponding new formulations should be evaluated.

In addition, because noisy data blur the signal of the leaf development process, (1) alternative data may be used, (2) observation protocols may be revised, and (3) visually observed data may be carefully selected. First, alternative data to calibrate and validate models of leaf senescence include data recorded with phenocams and remote sensed data in which leaf senescence dates are identified through the measured greenness, machine learning algorithms, and vegetation indices (Donnelly et al., 2022; Dronova and Taddeo, 2022; Gong et al., 2024; Richardson, 2023; Zeng et al., 2020). While these data are species-specific if recorded with phenocams, this may not be the case for remote sensed data (Joiner et al., 2016; Tang et al., 2016). Second, revised observation protocols should describe how to determine dates of leaf senescence stages (i.e., senescence induction, $LS_{50}$, and $LS_{100}$ at least) based on the measured, rather than estimated, state of leaf senescence. Such a measurement could be based on the greenness derived from images taken with consumer-grade digital cameras (Ide and Oguma, 2013; Richardson et al., 2018; Toomey et al., 2015; Zimmerman and Richardson, 2024). Moreover, a given observational time series should be based on at least 25 trees which are measured every other week (Liu et al., 2021; Morellato et al., 2010). Third, visually observed leaf senescence data should be selected primarily from the point of view of precision, for example by ensuring identical observation protocols and by sampling from cleaned data with a minimum of breakpoints. For this, the time series may be cleaned from outliers (Schaber et al., 2010) and separated into series without sudden changes in the mean (e.g., through a breakpoint analysis; Auchmann et al., 2018), before being sampled, preferably through spatially and climatologically stratified sampling, according to the research focus (e.g., gaining insight in the underlying processes or producing most accurate or most precise predictions Meier and Bigler, 2023).

## 5     Conclusion

The DP3 model builds on three subsequent phases of leaf development: the young, mature, and old leaf phase. The young leaf is insensitive to stress and transfers into a mature leaf solely due to aging. The mature leaf answers to aging and stress, both of which may induce senescence. While aging induces senescence with the transition from mature to old leaf, stress may already do so during the mature leaf phase through combining cold stress, photoperiod stress, and

585 dry stress. The output of the DP3 model includes daily rates of aging rates as well as of cold, photoperiod, and dry stress along with the dates of transition from young to mature to old leaf, senescence induction dates, and the leaf senescence dates. Thus, the DP3 model allows to develop testable hypotheses about the leaf senescence process, for example regarding the effect of site conditions on the timing of senescence induction and the duration of senescence as well as on the relative importance of cold, photoperiod, and dry stress. For example, the DP3 model predicted earlier senescence induction for warmer conditions due to aging and together with longer senescence, which may be tested through experi-

590 ments and in situ observations. This makes the DP3 model an important tool in the research of leaf senescence.

The accuracy of the DP3 model and of previous models of leaf senescence was lower than the accuracy of the Null model (i.e. the constant prediction of the average observation in the calibration sample). This was probably due to model formulations that do not fully reflect the leaf senescence process and, more importantly, to the leaf senescence data used for calibration and validation. Visually observed leaf senescence data are susceptible to observer bias and

595 based on observation protocols that are partly inconsistent between countries. Such noisy data blur the signal of the leaf senescence process, thereby probably forcing the models to resort to the average observation. This leads to low accuracy, regardless from the model formulation, which hinders the necessary further development of process-oriented models of leaf senescence.

The model error of the compared models was similarly affected by climatic and spatial deviations from the

600 calibration sample across models, and varied mainly due to the leaf senescence data. The similar effect of climatic and spatial deviations on the model error across models (including the Null model) illustrates that these models were heavily biased towards the mean. Moreover, the degree of noise in the used leaf senescence data is exemplified by these data accounting for 90% of the explained variance in the model error. Therefore, these data should be selected with particular attention to precision, e.g., by using as few sites with identical observation protocols as possible. Moreover, revised ob-

605 servation protocols should include senescence induction dates and rely on measurements rather than visual estimates. Such measurements may be based on the greenness of leaves to identify the degree of color change, involving digital cameras and automated image assessment.

**Code and data availability**

The R code for the DP3 model is openly available on Zenodo (Meier, 2025b, https://doi.org/10.5281/zenodo.14749339), together with the R code for the 2-phased version of the DP3 model ('DP2 model'), i.e., the DP3 model without young

leaf phase. While all raw data used are publicly available and referenced in section 2, the simulated leaf senescence data analyzed is openly accessible under https://doi.org/10.5061/dryad.tht76hf97 (Meier, 2025a).

**Author contributions**

MM, IC, and CB initialized the study and the model development. MM and IC conceptualized the final study and model development. MM designed the methodology and created the models, analyzed the models with input from IC and CB, visualized the results with input from IC, and wrote the draft with contributions from IC and CB. All authors approved

the final manuscript.

**Competing interests**

The authors declare that they have no conflict of interest.

### Acknowledgments

We thank the two anonymous reviewers and Shilong Ren from the Egusphere community for the critical reading of our manuscript and the helpful comments. Further acknowledgment is given to the E-OBS dataset and the data providers in the ECA&D project (https://www.ecad.eu). All leaf senescence models during model development and for model evaluation were calibrated and validated on the high performance computing cluster at the University of Lausanne. This study was funded by the Swiss National Science Foundation SNSF (project 210744).

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
