# Peer review of "Aging explains earlier start of leaf senescence in trees during warmer years: translating the latest findings on senescence regulation into the DP3 model (v1.1)"

_EGUsphere, 2025_

## Author Comment (AC1)

Dear Reviewer 2,

Thank you for your positive response to the initial submission of our manuscript. We are pleased to provide a strongly revised version, based on your comments and on those of reviewer 1 and of Shilong Ren (community comment 1), including our own further revisions. All changes are highlighted in yellow, whereas light yellow indicates shifted but unchanged text. Further, we provide our detailed responses to your comments here below (**bold** text). Note that references here below without DOI also appear in the manuscript, where they are listed including DOI.

Best regards,
Michael Meier, Christof Bigler, and Isabelle Chuine

* * *

Reviewer 2
(https://doi.org/10.5194/egusphere-2025-460-RC2)

General Comments

This study presents an interesting and innovative approach to modeling leaf senescence, which remains a challenging process to simulate. The work offers two key contributions:

   Unlike previous process-based models that focus solely on leaf senescence, the DP3 model attempts to represent the entire leaf development process from spring to autumn (that means from leaf unfolding to leaf senescence).
   The authors analyze the influence of leaf senescence data quality on model performance, which often overlooked in modeling studies.

While the proposed model introduces a novel structure, the results indicate that it does not yet simulate leaf senescence dynamics well. The authors attribute this primarily to data quality limitations. However, given the complexity of leaf development processes (from unfolding to senescence), model performance may also depend on how well these processes are represented. Phenology data derived from camera observations (e.g., PhenoCam) are less susceptible to observer bias and sampling uncertainty compared to traditional ground observations. Have the authors considered using such datasets to further evaluate the model structure?

**While we thoroughly discuss the representation of the processes in the model formulation (L404–418, L485–494, L521–530) as well as the uncertainty in visually recorded phenology data (L511–519, L531–541), we did not consider to reevaluate the model with new data from PhenoCams. Although this is a very compelling idea and would likely yield valuable results, we have no such data right now to try this and would be very interested in accompanying such a new study with researchers who would have extended enough time series of leaf senescence dates evaluated with images.**

The hypothesis that aging and stress drive leaf development is compelling. A discussion comparing this approach with more conventional growing-degree-day-based models would strengthen the manuscript. For instance, what are the advantages of using aging and stress as drivers instead of accumulative growing season temperature?

**We wrote a new paragraph to discuss this (L404–418) and tried to include this thought in the introduction to newly derived hypotheses from the DP3 model (L427) to discuss it later (L454–464).**

Interestingly, the results highlight cold, daylength, and dry stress as key drivers—similar to the Growing Season Index (GSI) model, which relies on minimum temperature, photoperiod, and vapor pressure deficit (VPD). Did the authors test VPD as an alternative drought stress indicator?

**Unfortunately, we did not test any alternative drought indices. However, we now discuss this shortcoming in section 4.4 (L545–550).**

Specific Comments

Line 43, 105: Please clarify the definition of leaf senescence. Autumn phenology typically distinguishes between leaf coloring and leaf fall as separate stages. How is senescence defined in this study based on both events?

**While we use leaf senescence as collective term for leaf coloring and leaf fall (now stated accordingly in L44–45), we based our study on the autumn phenology stages BBCH95 and BBCH97 for pome and stone fruit according to Meier (2018, https://doi.org/10.5073/20180906-074619) as now specified in L121–122.**

Figure 1C: The delayed leaf senescence at higher latitudes appears counterintuitive, as senescence usually occurs earlier in such regions. Could the authors provide insight into possible causes for this pattern?

**The regression through the function geom_smooth in the R package ggplot2 was calculated separately to each response variable (i.e., average $LS_{50}$ and average $LS_{100}$) as well as separately to each explanatory variable (latitude, longitude, and elevation. Thus, a positive relationship emerged between both $LS_{50}$ and $LS_{100}$ and latitude probably because the more northern sites are generally lower elevated. This is misleading, as you have pointed out, and we have corrected it accordingly. In the revised version of the manuscript, we fitted a linear regression to combined latitude, longitude, and elevation for each response variable (Sect. S1.1.2). These regressions indicate a negative relationship between the response variables and each explanatory variable. Figure 2c (former Figure 1c) was adjusted accordingly by plotting the results of these regressions for each explanatory variable, while keeping the other explanatory variables constant (i.e., set to the mean).**

Table 1:

Please include details on the spring phenology (LU, leaf unfolding) data.

**The additional information was included.**

The dataset combines observations from PEP725 and SPN. Were these collected using the same protocols? If not, how might protocols' differences affect model performance? Have the authors tested the model using only one dataset to assess potential improvements?

**We had no access to the precise protocols used to collect the data. As these protocols were established by different institutions from different countries, they likely differ (Menzel, 2013, https://doi.org/10.1007/978-94-007-6925-0_4). However, the same stages were visually observed among countries (i.e., the stages BBCH15, BBCH95, and BBCH97 according to Meier, 2018, https://doi.org/10.5073/20180906-074619). While we did not use data from only one country, the example script we provided together with the code for the DP3 model (Meier, 2025, https://doi.org/10.5281/zenodo.14749339) runs on data from only three sites. In consequence, the accuracy of the predictions for both the observations in the calibration sample and validation sample is considerably improved. This emphasizes our suggestion that the heavy noise in the used data blur the signal of leaf senescence. Thus, comparing the DP3 model with current models based on observations that do not contain any sudden changes in the mean (Auchmann et al., 2018) is an important way forward, which we now suggest in lines 576–580.**

Line 121: Please briefly describe the E-OBS dataset.

**We now do so (L139–141).**

Lines 124–125: Could the authors elaborate on the temperature correction method applied?

**We had done so in the lines 127–130 of the original manuscript, but probably did not emphasize this enough. We now restructured the paragraph a bit, such that the temperature correction method is now easily identified (L143–148).**

Line 135: Which remote sensing dataset was used?

**Here, we referred to the remote sensed $CO_2$ dataset. However, this was unclear, so we recited the dataset (L156).**

Line 139: Since LAI and $CO_2$ concentration are provided as monthly data, how was daily photosynthetic activity derived?

**These data were combined with daily values of surface shortwave down welling radiation, day length, and mean temperature. We now have clarified this in L160–161.**

Line 141: The Keetch-Byram Drought Index (KBDI) was selected as the drought metric. Were other indices (e.g., SPEI, PDSI) tested? If so, how did they compare?

**Unfortunately, we did not test any other drought indices. Considering the many drought indices there are, such a comparison would have inflated our manuscript too much. However, we totally agree with you that such a comparison would certainly be very valuable and believe that it would yield an entire study by itself. Nevertheless, we now briefly discuss this in section 4.4 (L545–550).**

Line 152: Does "several" refer to 34 sites? If so, please specify for clarity.

**Yes, in the end, we constructed and tested 34 formulations. However, this number of formulations is a result rather than a component of the method applied. Therefore, we mention it in the first line of the results section (L328). However, in order to avoid confusion, we simply omitted the word «several» in the method section (L173).**

Lines 200–204: The parameters *a* and b0 are set to 0.01 h. Could the authors justify this choice?

**These are examples. The calibrated values are listed in Table 3. We now made this clearer in the text (L224).**

Lines 218–219: Please define "extreme conditions" (e.g., hottest temperatures >30°C, coldest below a certain threshold).

**Rather than using a threshold to identify these conditions, we selected the site-years that contained the hottest 10-day period during the growing season observed in the dataset. We did so, because we wanted to select exactly 250 site-years. We have now specified this in lines 240–241.**

Figure 4 (3rd iteration): Does *f* represent h(x)? If so, please clarify in the caption.

**Yes, it does. We have now corrected this and revised the figure (now Fig. 6) completely.**

Table 2:

dm→o: Should this be interpreted as the simulated transition timing from mature to old leaves?

**Yes, it should. We have now corrected the definition (Table 2).**

SAging,I / SStress,i: Do these states accumulate since LU (leaf unfolding)?

**No, these states are the accumulated corresponding rates since the transition from young to mature leaf. We have now clarified this in the definition column of Table 2.**

Line 380: The authors associate cold stress with spring frost events. Could the importance of cold stress after midsummer also be examined?

**Not directly. An additional assessment would be necessary to examine cold stress accumulated before and after summer solstice. Rather than including such an additional assessment, we now included some results regarding the relative importance of cold stress during senescence (i.e., the period from senescence induction to leaf senescence; L372–376; Fig. S3; Tables S5–S8) and further discussed effects of cold stress (L454–478).**

Line 388: "This maybe" → Please revise for clarity (e.g., "This may be due to...").

**We modified the sentence as «This may be explained by unrealistic model formulations, poor model calibrations, and noisy data to drive and calibrate the models, all of which we discuss here below» (L480–484).**

Line 400: The sentence structure could be improved for readability.

**We adjusted the sentence structure (L497–500).**

---

## Author Comment (AC2)

Dear Reviewer 1,

Thank you for your detailed response to the initial submission of our manuscript. We are pleased to provide a strongly revised version, based on your comments and on those of reviewer 2 and of Shilong Ren (community comment 1), including our own further revisions. All changes are highlighted in yellow, whereas light yellow indicates shifted but unchanged text. Further, we provide our detailed responses to your comments here below (**bold** text). Note that references here below without DOI also appear in the manuscript, where they are listed including DOI.

Best regards,
Michael Meier, Christof Bigler, and Isabelle Chuine

\* \* \*

Reviewer 1
(https://egusphere.copernicus.org/preprints/egusphere-2025-460#RC1)

This manuscript introduces a new process-oriented leaf senescence model considering the three leaf development processes (young, mature, old). The new model doesn't seem to outperform the previous models or the Null model, most probably because the noise in the calibration and validation data is pushing the model simulation closer to the mean observation of the calibration sample.

Essentially, this manuscript highlights the need for modelers to consider the frequently overlooked uncertainty in underlying data, which could have profound implications and would be inspiring to be published here. Yet several main concerns remain.

**First**, the manuscript does not adequately address the DP3 model or its relevance to the conclusions. That is, the discussion and conclusions would remain unchanged even without the development of the advanced DP3 model. This would possibly weaken this research's merit for publication in this journal. This includes but not limited to:

- The introduction (L75-90 mainly) fails to convey the deficiencies of earlier models or justify the development of this new model. It is not apparent how the 3-phase model is an advancement.

    **To make the need of for our DP3 model clearer we have restructured the introduction (former L75-90 are now L46-62) as well as have inserted Figure 1 and an additional paragraph (L94-106).**
- Discussion for model accuracy and model error would be nice to focus more on DP3 model and provide more statistics.

    **Model accuracy and model error are evaluated and discussed based on a model comparison. Therefore, the DP3 model and the models used for the comparison have to be mentioned. As the DP3 model behaves like the other models (i.e., including the Null model) we only see two key messages here, which have been clearly stated: (1) the accuracy of the DP3 model is within a reasonable range, and (2) the model error mainly depends on data structure, which implies noisy data. Moreover, rather than introducing additional statistics, we decided to include the results regarding the senescence induction date (Fig. 7; L352–359).**

- As there is no improvement in accuracy perspective, would be necessary to show the advances in formulation. Yet section 4.1 lacks the comparison with the previous models, as well as the scientific evidence to support DP3 model findings. And absolutely lacks more implications from model development (see below).

  **We have now completed section 4.1 with that regard (L406–478).**
- Model accuracy and model error sessions seems to elaborate the same issue, might as well thinking about making them more concise.

  **The sessions elaborate different issues: Model accuracy focuses on the justification of the model, as probably both better and similar accuracy as previous models would justify the use of the DP3 model. Model error focuses on model behavior, illustrating that all the DP3 model as well as previous models behave similarly as the Null model. We hope to have clarified this through our revision (see below).**

**Second**, the DP3 model development resembles more like a data analysis exercise. It lacks a solid theoretical foundation or a comprehensive scientific interpretation of the model's outcomes.

- Regarding the DP3 model development, need to justify assumptions first by providing enough evidence and references. For example (if I understand correctly):
  - Stresses act as a compound event instead of several individual events to trigger leaf senescence.

    **Stressors act as individual events, but add up and accumulate as one (Eqs. 1 & 4). If this assumption is true or if each stressor accumulates individually, inducing senescence when either stressor-specific threshold was breached, is, according to out knowledge not known yet. However, because the referenced literature clearly mentions stress induced senescence in general rather than senescence induced by either cold stress or photoperiod stress in particular, we summed the stress events before accumulation.**
  - Legacy of stresses (all of them) accumulated from the very early spring on tree leaf senescence.

    **Current knowledge states that stress accumulates and may induce leaf senescence during the mature leaf phase (Fig. 1 in Jibran et al., 2013; https://doi.org/10.1007/s11103-013-0043-2). This assumption cannot be justified further and is based on all evidence we are aware of and which we have referenced (L46-63).**
  - Within leaf lifespan the relationship between age and stress effects remains unchanged for triggering senescence.

    **Non of the current studies (referenced in L46-63) implies a change in the relationship between aging and stress. Therefore, and by applying Occam's razor, we implemented the simpler formulation of a constant aging-stress relationship.**
  - Reasons for choosing the three main stresses (especially for dryness) and three additional stresses.

    **The reasons for our choice have been given (including references) in lines 46-63.**
- The discussion does not sufficiently cover the scientific importance of model selection, model formulation, or model parameter outcomes. In general, I would like to see more interpretations regarding them in the discussion part. I care about this because, as your

manuscript indicates that no matter how much improvements you make for the model structure, you would fail to 'predict'. Therefore, prediction accuracy might not be supposed to be the only goal. Might be necessary to focus more on what the development will bring us scientifically. If makes sense, might be interesting to know:

- What is the implication of 'more accurate' model. Does it really represent a model with better science? Or simply a model with less noise? This would be the foundation for the followings.

    **This is an interesting question. At first, a «more accurate» model seems to be a model that predicts more accurately. However, it may well be a model that is formulated more accurately, which could benefit predictions under changing climatic conditions. We believe that an accurate formulation is important for valid predictions as well as for the research of the processes that the model simulates.**

- The simpler the model [g(x), sudden response, rather than h(x), gradual response], the better the performance. Is it a victory for science or for statistics? Also, the case for 'product' outperforms 'exponential' function.

    **We are afraid, but feel that this question cannot be answered yet. While the true stress responses are likely gradual, steep stress gradients may well be approximated with sudden changes in stress. Moreover, because more parameters strengthen compensation effects between them, which we discuss in lines 441–446, responses with less parameters may yield a more stable model.**

- What is the implication for aging doesn't have much influence (better presented by 1) for senescence, which is a contrast to some previous research?

    **This likely is an artifact from the calibrated threshold for photoperiod stress, which results in stress of 1 being added on almost each day in the second half of senescence (i.e., the period between senescence induction and $LS_{100}$). Thus, photoperiod stress acts almost like an age count during senescence (Fig. S3; L373–376).**

- Additional factors not important for leaf senescence prediction, why?

    **While these additional factors were not important for leaf senescence prediction within the climate envelope used for model calibration and validation, they may be important within a wider climate envelope. Moreover, model selection based on more precise data with a clearer climate signal of leaf senescence may result in some of these additional factors being included (L558–575).**

- Three-phases model surpasses two-phases model, any implication?

    **The young leaf phase, which does not answer to stress, becomes important as soon as the state of senescence must follow a path laid out by at least two stages of leaf senescence. This exemplifies the need of such data (L564–566).**

- Yet in table 3 it shows that the best $Y_{Aging,1}$ is only 1.57d. It is pretty short that the new model can basically regarded as a 2-stage model, which undermines the formulation test from the second iteration. Might be helpful if there is a sensitivity plot. Also I wonder if mature leaf span of around 70 days is realistic?

    **The original manuscript listed the parameters of the DP3 model calibrated with the $LS_{50}$ sample in Table 3. However, model selection was based**

on the $LS_{50}$-$LS_{100}$ sample. We now included the parameter of both models in Table 3, and those of the DP3 model calibrated with the $LS_{50}$-$LS_{100}$ sample appear more realistic. While this illustrates the compensation effect mentioned above (L441–446), it also shows the need for data that contains more than one stage of leaf senescence (L564–566).

- Downside of this model.

    We now discuss how the DP3 model could be improved in section 4.4 (L543–557).

In summary, we have now discussed the DP3 model thoroughly in section 4.1 (L404-478), together with way to improve the model in section 4.4 (L543–557).

Specific comments:

L1: There is a lack of 'the latest findings' for this research, or model development, also doesn't appear to be the focus of your research. Might be a better title if concentrate more on data quality.

**While we have changed the title (L1–3), the latest findings have been summarized in the introduction (former L75-90, now L46-62) and are now illustrated with figure 1.**

L30-34: Please be careful about the suggestion of using 'as few sites as possible' as you don't remind the difficulties of application at larger scales. And the last sentence is pretty hard to grasp the meaning.

**We have rephrased our suggestion (L32-34).**

L40: 'nutrient resorption' instead of 'nutrient retraction'?

**We have changed this accordingly (L41).**

L90: Yet what is the problem with the current progress to draw you developing this new model, instead of testing the existing models?

**To clarify the need of for our DP3 model, we have restructured the introduction (former L75-90 are now L46-62) as well as have inserted figure 1 and an additional paragraph (L94-106).**

L97: Confused by the exact meaning of 'relationship'. And not mentioned in the introduction session.

**We have rephrased research question 1 (L113).**

L105: Please indicate the phase id if possible.

**We have done so in L121-122.**

L132: I wonder why not taking one dataset as a reference and correct two CO2 datasets at a same level? This might bring a change to year 2013-2022.

**We have chosen this procedure as we have been unable to decide, which dataset should be used as reference. Also, we are confident that the here applied procedure did not add an artifact to the calculated $A_{net}$ that drives the PIA model (this was our only use of the $CO_2$ data; see the figure here below for ten randomly selected sites).**

[Figure]

L135: I am a bit confused by what missing observation (variable) you mean here? And what do you mean by 'weighted' for the average LAI?

**We clarified, which modeled data was used (L154–155) and what was meant by weighted average LAI, which is now explained at the very beginning of section 2.2 (L137–139).**

L140: How do you calculate the 'day length, daily photosynthetic activity…' in absence of LAI for the past (1950-1981 at least)?

**We have clarified this in L149–150.**

L180, 185: what is the difference between the definitions of 'cold days' and 'frost days'?

**We have clarified the difference in the text (L206 and through the new Table S3).**

L184: The seasonal cycles of nutrient depletion will be represented by LAI. Yet I wonder if LAI would be a proper metric here if spectrum products would reflect more directly the nutrient supplies for the plants.

**While LAI represents the seasonal cycle of photosynthetic productivity ($A_{net}$; see Sect. S1.2.2), nutrient depletion is modeled as a function of the accumulated $A_{net}$ since the day of leaf unfolding. This is of course a rough approximation, which we have used in the absence of any better suited (soil) data.**

L262: what is 'cold degree-days day length'

**We have corrected this typo (L287).**

L230-240: Model development part is a bit hard to follow. It would be far easier to follow this part if you could relate the texts with a figure similar to Fig. 4.

**We have inserted Figure 5.**

L232-234: The definition of the senescence rate is quite hard to understand. Would you please rephrase it into short sentences maybe?

**We have rephrased and split the sentence (L254-257).**

L234-235: The manuscript doesn't present the 2-phase development settings, which leaves me puzzled about the justification of your iteration design. Please adding more details about the 2-phase model in section 2.3, ideally, at least, with a adding figure like figure 2, and an explanation of the 2-phase model's structure.

**We have done so in the first paragraph of section 2.3 (L164-171) and in Figure 3 (panels b and d).**

Also, please provide the source code or at least instructions for the 2-phase model implementation in the DP3 model source code.

**We have provided the corresponding code and mentioned so in line 605–606.**

L239-240: Hard to understand the 'subsequent iterations' settings.

**We changed "In subsequent iterations" to "In iteration 6" make the phrase easier to understand (now L263–264). Moreover, we included Figure 5 to illustrate the procedure.**

L256-257: Including analysis along ELV considering the dry stress remains valid here. Yet it shows in Fig. 4 that the nutrient item has been omitted, and Fig. 5 clearly associates ELV more with photoperiod stress. Could you please provide more insights into this mismatch between your assumption and your findings in the discussion?

**We have elaborated these issues in sections 4.1 and 4.4 (L404-478 and L543-557).**

Fig. 4:

**We have revised Figure 4 (now Figure 6) to clarify the development of the DP3 model. Your corresponding comments have been answered specifically here below.**

- Could be related to previous comments. For the 1$^{st}$ iteration, please explain more for the setting of 2 phase model. And why here choose to go with 2-phase instead of directly 3-phase if you wouldn't add more discussions later.

    **We started with the 2 phase model, which is now clearly illustrated in Figures 5 & 6. The setting of the 2 phase model is explained in Figure 3 and in lines 165-166.**
- For the 4$^{th}$ iteration results, what is 3_D_gCgLgDdP_P? Is it 3_D_gCgLgDgP_P.

    **Yes, it is. This has been corrected (now Fig. S1).**
- For the 5$^{th}$ iteration tested formulations, what is '…fP_P'? Why not testing hP?

    **This was a typo and we meant hP, which we have now corrected (now Fig. S1).**
- For the 6$^{th}$ iteration results, same as above.

    **This was the same typo, which we have now corrected (now Fig. S1).**
- For the 6$^{th}$ iteration, does it mean you are adding the additional factors one by one until you test all factor combinations from 3 factors to 7 factors? If so, show it clearly in the figure. If not, please describe more clearly how you did it in the method session, and explain why you are not testing all 7 factor combinations together.

    **We started with three stressors (i.e., the most probable stressors cold days, shortening days, and dry days) and continued with a forward selection of additional stressors, always considering each stressor through the response functions *g(x)* and**

***h(x)*. We have clarified this by revising section 2.5 (L251-271) and the revised Figure 6 as well as by including the new Figure 5.**

Table 3: If possible, would you please add in the note that which day '-0.0016h' corresponds to?

**We have done so in Table 3 as well as mentioned the corresponding dates in the not to the table.**

Table S3: What does 'both' mean? Like exactly the same date for the causes of stress and aging to happen? I wonder if you could visualize the table by histograms to show how causes act differently regarding the variables you consider?

**Here, 'Both' refers to the site-years during which aging and stress reached their thresholds on the same date. We have clarified this in the table (now Tables S5-S8). To illustrate these causes better, we have revised Figure 7 (former Figure 5).**

L334: How much is the importance of the dry stress? At least show the results in the supplementary materials please.

**We have now mentioned this in the text (L 367) and in Tables S5-S8.**

Fig. 5a: Could you please label the number of site years for each sample?

**We have altered the figure (now Figure 7) and visualized the number of site-years in the top row of each panel.**

Table 4: 'P spatial'

**We have corrected this.**

L245: Would you please compare with the other research so we could see if the model error is within a normal range?

**We are unaware of any study the evaluated the model error for stages of leaf phenology with one exception: Meier et al. (2023; https://doi.org/10.1111/gcb.17099) assessed the strength of the bias to the mean in leaf senescence models through the relationship between model error and phenological difference (i.e., the difference between a given observed date of leaf senescence and the average leaf senescence in the calibration sample). Thus, they showed that the model error depended strongly on this difference. In other words, the model error may be positive or negative and the absolute model error may be large or small, depending on the phenological difference. In consequence, the mean model error depends on the mean phenological difference, which varies between calibration and validation sample pairs and thus between studies. Therefore, to effectively compare the model error between studies, we would have to know the phenological difference between the calibration and validation sample in each study, which we do not.**

L351: What does 'evidence' mean?

**Evidence refers to the signal in the data against the null hypothesis, i.e., an effect of zero. Here, we have used the minimum Bayesian factor, which expresses the most optimistic change of odds between the null hypothesis and alternative hypothesis together with the *p*-value. This has been explained in Sect. 2.7, L320–323). In addition, we have reformulated the sentence in the former line 351 (now L393–397) to make this clearer.**

L365: As a quite important application, would you please give more details about it? Also the ecological importance of the timing transiting from mature to old leaf.

**We have revised section 4.1 completely (L404–478).**

L379-384: Can't understand the logical link between the two sentences. And please make it clear if it is cold stress or frost events (frost stress).

**This has been rephrased (L457–464).**

L390-L409: This part seems to lack a focus on the DP3 model. Would you consider delving deeper into the distinctions between your model and the other models more thoroughly?

**The other models were introduced in section 2.6 (L284–295). Here, we rather focused on the match between the used model and the research question / task. We have now clarified this by altering the paragraph (L480–494).**

L421: It is such a big gap from the proceeding sentence to this one, quite hard to follow the reasoning here.

**We have rephrased the two sentences (L517–519).**

L426-429: Would you elaborate on the explanations and implications of this phenomenon?

**We referenced to the possible reasons, namely unrealistic model formulations, poor model calibrations, and noisy data, all of which were previously discussed in section S4.2). In addition, we referenced to Meier et al. (2023), who focused their study on this phenomenon. This has now been clarified through rephrasing (L523–530).**

L455-458: I don't quite follow your suggestion for the 'revised observation protocols'. Would you please rephrase and make it clearer how to implement the new protocols?

**We have rephrased this part (L564–569).**

L459: It might need to be more careful in conveying the usage of 'as few as possible' sites for model development. Instead, I would like to know more possibilities if we could select sites with the help of current knowledge of inter-annual and inter-site variabilities.

**We have extended our discussion of this subject (L569–575).**

And considering that model calibration/validation and evaluation are distinct aspects of dataset application, I wonder if treating these two parts of the dataset separately (since involving different tools) could lead to improvements in this field? Also, could provide different insights for model accuracy and error assessments?

**This is a very interesting thought indeed. As Meier et al. (2023; https://doi.org/10.1111/gcb.17099) have shown, the RMSE depends also on the choice of validation sample. Therefore, to enable inter-study comparisons, the validation samples should be selected as thoughtfully and sorrowfully as calibration samples. Again, this selection should be based on the research question the study focuses on, yielding different samples for, say, a study of the underlying process versus accurate projections under scenarios of future climate (L571–575).**

L462: Please specify 'three subsequent phases'.

**The phases have been specified (L576).**

L462: Please include more scientific indications from the DP3 model, in addition to the 'structural strength' of this model.

**We have done so in lines 582–586.**

L484-486: Same as the above-mentioned comment – this outlook is unclear to me.

**We have clarified our idea of revised protocols (L600–603).**

---

## Author Comment (AC3)

Dear Shilong Ren,

Thank you for your community comment 1 and your interesting question to the initial submission of our manuscript. We are pleased to provide a strongly revised version, based on your comment and on the comments of the reviewers 1 and 2. All changes are highlighted in yellow, whereas light yellow indicates shifted but unchanged text. Please find our detailed responses to your comment here below (**bold** text).

Best regards,
Michael Meier, Christof Bigler, and Isabelle Chuine

* * *

Community comment 1
(https://doi.org/10.5194/egusphere-2025-460-CC1)

The authors developed a new and systematic model of the senescence process of woody plants and analyzed the impact of data quality on the simulation of autumn phenology. The research perspective is unique. After previous revisions, I think there are no issues with the article's structure and writing quality. I have only one question: According to the multi-model comparison results in the article, it is not difficult to find that the DP3 model does not outperform the PAI and DM2 models in terms of simulation accuracy. It seems that the improvement significance of the new model is not significant. Could it be that the model process is too complex and has too many parameters?

**We have now included the results for the DP3 model calibrated with one and with two stages of leaf senescence (i.e., with the stage when 50% of the leaves have turned color or fallen, $LS_{50}$, as well as with $LS_{50}$ and the stage when 100% of the leaves have turned color or fallen, $LS_{100}$). These two calibrations led to different results. For example, the young leaf phase lasts 41 days when calibrated with $LS_{50}$ and $LS_{100}$ but only 1 day when calibrated with $LS_{50}$ only. This illustrates a compensating effect between the different model parameters (i.e., different parameter sets yield similar results), which we now discuss in L441–446). However, while the probability for such compensating effects arguably increases with the number of parameters, it should be irrelevant for the accuracy of the predictions, provided that these remain in the space of the calibration conditions. This accuracy may only suffer by a high number of parameters when the calibration algorithm cannot handle them. Here, we selected the algorithm generalized simulated annealing, which has been used to successfully calibrate models with up to 30 parameters (Xiang et al. 2013, https://doi.org/10.32614/RJ-2013-002). Moreover, we carefully tuned this algorithm to the models (Sect. S2.2), why we are confident that the complexity of the model did not have any adverse effects on the accuracy of the predictions.**

---

## Author Response (AR2)

Michael Meier
michael.meier.1@unil.ch

Carlos Sierra
csierra@bgc-jena.mpg.de

Lausanne, 13.08.2025

**To the handling editor – tracked changes**

Dear Mr. Sierra,

I am happy to upload all documents for publication. When rereading our manuscript, my co-authors and I felt that some passages could be improved for easier understanding. In addition, we felt that consistency in the usage of the terms ‹simulation› and ‹prediction› should be improved. These improvements have been made and marked with tracked changes in the following main part and supplement of our manuscript.

I hope you may accept these changes and the we can proceed towards the final publication of our study. In the meantime, please do not hesitate if you may need any further information or have any questions.

Best regards,
Michael Meier

[revised manuscript text omitted]

We related the average leaf senescence date per site to latitude, longitude, and elevation through linear regression models fitted separately for the average day of year when 50% and 100% of the leaves have turned color or have fallen (average LS50 and average LS100, respectively; Eq. S7; using the function lm in the R package stats; R Core Team, 2025):

$$\mathbf{y} = \mathbf{X}\boldsymbol{\beta} + \boldsymbol{\epsilon} \tag{S7}$$

$\mathbf{y}$ is the $n_j$-dimensional vector of the response variables average $LS_{50}$ and average $LS_{100}$, where $j$ refers to either response variable and $n_j$ is the corresponding number of observations. $\mathbf{X}$ is the $n_j \times 4$ matrix with the values of the first column being set to 1 (for the intercept) and the $2^{nd}$ to $4^{th}$ columns containing the respective explanatory variables latitude [°], longitude [°], and elevation [m a.s.l.]. $\boldsymbol{\epsilon}$ is the $n_j$-dimensional vector of normally distributed errors with $N(0, \sigma_j^2)$. Thus, $\boldsymbol{\beta}$ is the 4-dimensional vector of the coefficient estimates for the intercept, latitude, longitude, and elevation.

The linear regression models revealed earlier average leaf senescence dates (i.e., $aLS_{50}$ and $aLS_{100}$) with increasing latitude, increasing longitude, and increasing elevation (Table S1). Inline with recent research, this translates into earlier leaf senescence with cooler and more continental climatic conditions as well as with longer days during summer and faster decrease in day length between summer solstice and winter solstice (e.g., Kloos et al., 2024; Wang et al., 2022; but see Lu and Keenan, 2022).

**Table S1. Coefficient estimates of the linear regression models**

| Response | Explanatory | Estimate | Standard error | *t* statistic | *p*-value | Adjusted $R^2$ |
|---|---|---|---|---|---|---|
| average $LS_{50}$ [doy] | Intercept | 386.35 | 21.13 | 18.2837 | 0.000000 | 0.4761 |
| | Latitude [°] | −1.69 | 0.40 | -4.2079 | 0.000036 | |
| | Longitude [°] | −1.61 | 0.15 | -10.9693 | 0.000000 | |
| | Elevation [m a.s.l.] | −0.0142 | 0.0031 | -4.5542 | 0.000008 | |
| average $LS_{100}$ [doy] | Intercept | 374.87 | 36.49 | 10.2731 | 0.000000 | 0.7614 |
| | Latitude [°] | −0.95 | 0.70 | -1.3631 | 0.175860 | |
| | Longitude [°] | −1.62 | 0.27 | -5.9722 | 0.000000 | |
| | Elevation [m a.s.l.] | −0.0219 | 0.0049 | -4.4497 | 0.000022 | |

*Note*: The linear regression models were fitted separately to the response variables average $LS_{50}$ and average $LS_{100}$ (i.e., the day of year [doy] when respective 50% and 100% of the leaves have turned color or have fallen). The coefficient estimates of the explanatory variables are given together with the corresponding standard errors, *t* statistic, and *p*-values. The adjusted $R^2$ is given for each model.

**S1.2    Driver calculations**

**S1.2.1    Day length**

Day length ($L_{doy}$) for a given day of year (*doy* [d]) was calculated from latitude ($\varphi$; [°]) according to Eqs. 1, 3, and 4 in Brock (1981; Eqs. S8–S10):

$$L_{doy} = 2 \frac{W_{doy}}{15° \, \mathrm{h}^{-1}} \tag{S8}$$

$$W_{doy} = \arccos\left(-\tan(\varphi) * \tan(\gamma_{doy})\right) \tag{S9}$$

$$\gamma_{doy} = 23.45° \sin\left(360° (doy - 81)/365\right) \tag{S10}$$

With $\gamma_{doy}$ and $W_{doy}$ being the respective declination [°] and hour-angle [°] at sunrise at *doy*.

**S1.2.2 Photosynthetic activity**

Sink limited daily net photosynthetic activity ($A_{net}$ [mol C d$^{-1}$]; Eq. S11; Collatz et al., 1991) was calculated as the difference between the gross photosynthetic activity ($A_{grs}$ [mol C d$^{-1}$]) and respiration ($R$ [mol C d$^{-1}$]; Collatz et al., 1991; Farquhar et al., 1980; Wohlfahrt and Gu, 2015).

$$A_{net} = A_{grs} - R \tag{S11}$$

$A_{grs}$ in turn depended on photon availability ($J_E$ [mol C d$^{-1}$]), Rubisco activity ($J_C$ [mol C d$^{-1}$]), and sink capacity ($J_S$ [mol C d$^{-1}$]; Eq. S12), while $R$ was defined as a fraction of the maximum photosynthetic rate ($V_{max}$ [mol C d$^{-1}$]; Eq. S13).

$$A_{grs} = \max\left(0, \quad L \times \frac{J_P + J_S - \sqrt{(J_P + J_S)^2 - 4\beta_C J_P J_S}}{2\beta_C}\right) \tag{S12}$$

$$R = b_{C3} V_{max} \tag{S13}$$

$J_P$ is an intermediate variable, combining $J_E$ and $J_C$ (Eq. S14), $\beta_C$ is a constant shape parameter, and $b_{C3}$ is a constant fraction for C3 plants (Table S1).

$$J_P = \frac{J_C + J_E - \sqrt{(J_C + J_E)^2 - 4\theta_C J_E J_C}}{2\theta_C} \tag{S14}$$

$\theta_C$ is a constant shape parameter (Table S1). $J_E$ and $J_C$ are daily fractions of the available photosynthetically active radiation (*APAR* [W m$^{-2}$]; Eq. S15) and $V_{max}$ (Eq. S16), respectively, while $J_S$ is a constant fraction of $V_{max}$ (Eq. S17).

$$J_E = C_1 \times \frac{APAR}{L} \tag{S15}$$

$$J_C = C_2 \times \frac{V_{max}}{24[h]} \tag{S16}$$

$$J_S = 0.5 \times \frac{V_{max}}{24[h]} \tag{S17}$$

$L$ is the day length [h], and $V_{max}$ depends on *APAR* (Eq. S18), which in turn was calculated as a fraction (*fapar*) of the photosynthetically active radiation (*PAR* [W m$^{-2}$]; Eq. S19).

$$V_{max} = \frac{1}{b_{C3}} \frac{C_1}{C_2} [(2\theta - 1)s - \sigma(2\theta s - C_2)] APAR \tag{S18}$$

$$APAR = \alpha_a\, c_q\, fapar\, PAR\, (3600 \times 24)\,[\text{s}] \tag{S19}$$

$\theta$ is a constant shape parameter, while $\alpha_a$ and $c_q$ are a constant ratio and a constant conversion factor for the respective assimilation and conversion of solar radiation (Table S1). While *fapar* depended on the leaf area index (*LAI*; Eq. S20), *PAR* was derived from the surface shortwave down welling radiation ($R_s$ [W m$^{-2}$]; Eq. S21).

$$fapar = 1\, e^{-0.5\, LAI} \tag{S20}$$

$$PAR = 0.5\, R_s \tag{S21}$$

$V_{max}$ further depends on $s$ and $\sigma$ (Eqs. S22–S23) as well as on $C_1$ and $C_2$ (Eqs. S24–S25).

$$s = b_{C3}\, \frac{24\,[\text{h}]}{L} \tag{S22}$$

$$\sigma = \sqrt{1 - \frac{C_2 - s}{C_2 - \theta s}} \tag{S23}$$

$$C_1 = \phi_C\, \alpha_{C3}\, f(T) \times \frac{p_{i,CO2} - \Gamma_*}{p_{i,CO2} + 2\Gamma^*} \tag{S24}$$

$$C_2 = \frac{(p_{i,CO2} - \Gamma_*)}{p_{i,CO2} + K_C\left(1 + \frac{p_{a,O2}}{K_O}\right)} \tag{S25}$$

$\alpha_{C3}$ describes the quantum efficiency of C3 plants, and $p_{a,O2}$ is the ambient partial $O_2$ pressure (Table S1). $p_{i,CO2}$ is the internal partial $CO_2$ pressure (Eq. S26), $\Gamma_*$ is the $CO_2$ condensation point (Eq. S27), $K_C$ and $K_O$ are the kinetic coefficients for $CO_2$ (Eq. S28) and $O_2$ (Eq. S29), respectively, and *f(T)* is a function of the mean temperature (Eq. S30).

$$p_{i,CO2} = \lambda_{C3}\,[CO_2]_A\, 10^{-16}\, p_0 \tag{S26}$$

$$\Gamma_* = \frac{p_{a,O2}}{2\tau\, q_{\tau 10}^{(T - 25\,K)/10}} \tag{S27}$$

$$K_C = k_C\, q_{C10}^{(T - 25\,K)/10} \tag{S28}$$

$$K_O = k_O\, q_{O10}^{(T - 25\,K)/10} \tag{S29}$$

$$f(x) = \min\left(1, \quad \max\left(0, \quad \frac{1}{1 + e^{k_1(k_2 - T)}} \times \left(1 - 0.01\,e^{k_3(T - x_3)}\right)\right)\right) \tag{S30}$$

[revised manuscript text omitted]

**S2    Methods**

**S2.1        Initial ranges for parameters**

**Table S3. Ranges for parameter calibration**

| Symbol | Meaning | Boundaries |
|---|---|---|
| $-a_C$ | Boundary below which cold stress is 1 versus 0 | 0–30 °C |
| $b_{0,C}$ | Boundary above which cold stress gradually increases from 0 to 1 | 0–30 °C |
| $b_{1,C}$ | Boundary below which cold stress gradually decreases from 1 to 1 | $(b_{0,C}+0$ °C)–$(b_{0,C}+20$ °C) |
| $-a_P$ | Boundary below which photoperiod stress is 1 versus 0 | –0.25–+0.25 h |
| $b_{0,P}$ | Boundary above which photoperiod stress gradually increases from 0 to 1 | –0.25–+0.25 h |
| $b_{1,P}$ | Boundary below which photoperiod stress gradually decreases from 1 to 0 | $(b_{0,P}+0$ h)–$(b_{0,P}+0.3$ h) |
| $a_D$ | Boundary above which dry stress is 1 versus 0 | 0–800 |
| $-b_{0,D}$ | Boundary above which dry stress gradually increases from 0 to 1 | 0–800 |
| $-b_{1,D}$ | Boundary below which dry stress gradually decreases from 1 to 0 | $(b_{0,D}+0)$–$(b_{0,D}+400)$ |
| $a_R$ | Boundary above which rain stress is 1 versus 0 | 0–500 mm |
| $-b_{0,R}$ | Boundary above which rain stress gradually increases from 0 to 1 | 0–500 mm |
| $-b_{1,R}$ | Boundary below which rain stress gradually decreases from 1 to 0 | $(b_{0,R}+0$ mm)–$(b_{0,R}+300$ mm) |
| $a_H$ | Boundary above which heat stress is 1 versus 0 | 25–50 °C |
| $a_N$ | Boundary above which nutrient stress is 1 versus 0 | 20–250 mol C d$^{-1}$ |
| $-a_F$ | Boundary below which frost stress is 1 versus 0 | –5–+10 °C |
| $w_C$ | Weight of cold stress | 0–1 |
| $w_P$ | Weight of photoperiod stress | 0–1 |
| $w_D$ | Weight of dry stress | 0–1 |
| $w_R$ | Weight of rain stress | 0–1 |
| $w_H$ | Weight of heat stress | 0–1 |
| $w_N$ | Weight of nutrient stress | 0–1 |
| $w_F$ | Weight of frost stress | 0–1 |
| $w_A$ | Weight of aging rate | 0–1 |
| $w_S$ | Weight of stress rate | 0–1 |
| $s_X$ | Scaling factor of the senescence rate | 0–1 |
| $x_S$ | Shape parameter of the stress rate | 0–10 |
| $c$ | First parameter of exponential function | 0.005–0.5 |
| $d$ | Second parameter of exponential function | 0–15 |
| $Y_{Aging,1}$ | Threshold for the aging state, marking the transition from young to mature leaf | 0–50 d |
| $Y_{Aging,2}$ | Threshold for the aging state, marking the transition from mature to old leaf | $(Y_{Aging,1}$$)$–$(Y_{Aging,1}+250$ d) |
| $Y_{LS100}$ | Threshold for the senescence state, indicating the day of LS$_{100}$ | 0–10 |

*Note*: The symbols of the parameters for the boundaries below or above which stress occurs [$a$; see response function $g(x)$; Eq. 7], for the boundaries between which stress occurs [$b_0$ and $b_1$; see response function $h(x)$; Eq. 8], and for the weights ($w$) that define the stress rate as well as for the different formulations of the senescence rate ($w_A$, $w_S$, $s_X$, $x_S$, $c$, and $d$; Eq. 9) and the thresholds ($Y$) that mark the transitions from young to mature leaf, the transition from mature to old leaf and the time when 100% of the leaves have changed color or have fallen (LS$_{100}$).

**S2.2        Controls of the simulated annealing algorithm**

The choice of the controls for the optimization algorithm influences the accuracy of the calibrated model (Meier and Bigler, 2023) through the exploration–exploitation trade-off (Candelieri, 2021; Maes et al., 2013). Thus, we set the controls 'maximum iterations', 'maximum calls', and 'temperature' of the generalized simulated annealing algorithm (Xiang et al., 1997, 2017) in such a way that the calibrated model resulted in most accurate predictions for the

validation sample. To identify these optimal controls for each model and calibration sample, we calibrated each model four times (i.e., twice with each sample draw) with all 27 combinations of 4000, 5000, and 6000 maximum iterations, $10^6$, $10^7$, and $10^8$ maximum calls, as well as temperatures of 5200, 5230, and 5300. Thus, we used the combination of controls that resulted in the lowest average Akaike information criterion for small samples (i.e., $n < 40k$; AICc; Eq. S41; based on the validation sample; Akaike, 1974; Burnham and Anderson, 2004) to compute the additional six calibration runs (i.e., three per calibration sample; Table S4).

$$AICc = AIC + \frac{2\,k\,(k+1)}{n-k-1} \tag{S41}$$

$$AIC = -2 \times \log(L) + 2\,k \tag{S42}$$

$$\sigma_e = \sqrt{\frac{1}{n} \sum_{i=1}^{n} (x_{p,i} - x_{o,i})^2} \tag{S43}$$

$n$ is the number of predicted and observed  doy pairs ($x_{p}$ and $x_{o}$, respectively) and $k$ is the number of free model parameters. $L$ is the likelihood for the normally distributed model errors (i.e., $\boldsymbol{x_p - x_o}$; Fisher and Russell, 1997) with $N(0, \sigma_e)$. In case $S_{Senescence}$ did not reach the thresholds $Y_{LS_{50}}$ and $Y_{LS_{100}}$ until December 31$^{st}$, corresponding $x_{p}$ were considered missing and thus set to doy 367 before their accuracy was evaluated.

**Table S4. Optimal controls of the generalized simulated annealing algorithm.**

| Model | Sample | Maximum iterations | Maximum calls | Temperature |
|-------|--------|--------------------|--------------|-------------|
| CDD | $LS_{50}$ | 4000 | $10^8$ | 5300 |
| DM2 | $LS_{50}$ | 6000 | $10^6$ | 5300 |
| PIA | $LS_{50}$ | 5000 | $10^7$ | 5200 |
| DP3 | $LS_{50}$ | 4000 | $10^8$ | 5300 |
|  | $LS_{50}$-$LS_{100}$ | 5000 | $10^7$ | 5200 |

*Note*: Only the control settings for the evaluated models ($LS_{50}$ sample) and for the model that was selected through the iterations of model development ($LS_{50}$-$LS_{100}$ sample) are shown. Those for the models that were rejected during model development are omitted.

**S2.3      Model calibration, selection, and evaluation**

All models were calibrated by minimizing the root mean squared error (RMSE; Eq. S44).

$$RMSE = \sqrt{\frac{1}{n} \sum_{i=1}^{n} (x_{p,i} - x_{o,i})^2} \tag{S44}$$

Thus, for each model, we selected and further evaluated the calibration run that resulted in highest modified Kling-Gupta efficiency (KGE'; Eq. S45; Gupta et al., 2009; Kling et al., 2012) for the validation sample.

$$KGE' = 1 - \sqrt{(\rho-1)^2 + (\beta-1)^2 - (\gamma-1)^2} \tag{S45}$$

$$\beta = {}^{\mu_p}\!\big/\!{}_{\mu_o} \tag{S46}$$

$$\gamma = \frac{\sigma_p/\mu_p}{\sigma_o/\mu_o} \tag{S47}$$

$\beta$ is the bias ratio, $\gamma$ is the variability ratio, and $\rho$ is the Pearson correlation between $x_{sp}$ and $x_{so}$. $\mu_{sp}$ and $\mu_{so}$, are the respective predicted and observed  mean doy, and $\sigma_{sp}$ and $\sigma_{so}$ are the corresponding standard deviations. For the perfect model (i.e., $x_{sp} = x_{so}$ for all $i$), $\rho = 1$, $\beta = 1$, and $\gamma = 1$, and thus $KGE' = 1$, whereas $1 > KGE' > -\infty$ for imperfect models.

**S2.4    Linear mixed-effects model and analysis of variance**

We fitted a linear mixed-effects model (LMM; Eq. S48; Pinheiro and Bates, 2000; Wood, 2011, 2017) to analyze the effects on the model error:

$$\mathbf{y} = \mathbf{X}\boldsymbol{\beta} + \mathbf{Z}\mathbf{b} + \boldsymbol{\epsilon} \tag{S48}$$

$\mathbf{y}$ is the $n$-dimensional vector of the response variable 'model error' (ME) and $n$ is the corresponding number of ME. $\mathbf{X}$ is the $n \times p$ matrix of the intercept (i.e., 1) and the $p - 1$ explanatory variables. $\boldsymbol{\beta}$ is the corresponding $p$-dimensional vector of the fixed effects 'country' and 'model' as well as the annual and site-specific deviations in mean annual temperature, mean annual KBDI, accumulated $A_{net}$ between LU and summer solstice, latitude, and elevation (CTR, MOD, δMAT, δMAQ, δ$A_{net}$, δLAT and δELV, respectively) from the overall calibration sample means per variable. $\mathbf{Z}$ is the $n \times q$ matrix of the random effects, assigning the $n$ observations to the $q$ groups of the grouping variable 'site' (STE). $\mathbf{b}$ is the corresponding $q$-dimensional vector of the random intercepts with $\mathbf{b} \sim N(0, \sigma_\mathbf{b}^2 \mathbf{I}_q)$, and $\boldsymbol{\epsilon}$ is the $n$-dimensional vector of the errors with $\boldsymbol{\epsilon} \sim N(0, \sigma^2 \mathbf{I}_n)$ (Baayen et al., 2008; Chpt. 2.1 in Pinheiro and Bates, 2000; Chpt. 6.2 in Wood, 2017).

We fitted this LMM with the function bam in the R package mgcv (Wood, 2017), using the following formula (Eq. S49):

```
ME ~ MOD * (δMAT + δMAQ + δAnet + δLAT + δELV) + CTR + s(STE, bs = 're')    (S49)
```

This LMM combined model effects interacting with effects due to climatic deviations from the calibration sample (red), spatial deviations from the calibration sample (green), and data structure (blue). The LMM was the basis for the type-III ANOVA (Yates, 1934), which we derived with the functions aov and drop1 in the R package stats (Eq. S50; R Core Team, 2025):

```
drop1(aov(LMM), scope = ~., test = "F")    (S50)
```

Thus, we calculated the amount of variation attributed to differences among each explanatory variable, i.e., the relative impact of given variable on the variance in the model error explained by the LMM, by dividing the variable-specific sum of squares by the total sum of squares over all variables.

**S3   Results**

**S3.1        Formulation of the leaf development process**

[Figure]

**Figure S1. Tested models.** The tested models were labeled according to their formulation, namely as $x_P\_x_A\_x_S\_x_X$, with $x_P$ being the number of leaf development phases (i.e., 2 or 3), $x_A$ being the driver of the aging rate (i.e., A or D for photosynthesis or days, respectively), $x_S$ being the stress rate in response [i.e., g or h for $g(x)$ or $h(x)$] to the stressors cold (C), shortening (P), dry (D), heat (H), and frost (F) days, heavy rain periods (R), and nutrient depletion (N), and $x_X$ indicating the formulation of the senescence rate (i.e., S, P, or X when formulated as a sum, product, or exponential function of aging and stress, respectively). After each iteration, we identified the two most accurate models across the given and all previous iterations (Fig, 5, Sect. 2.5). These models were further developed through the next iteration. As soon as such a subsequent iteration did not produce any new model, we selected the most accurately formulated model among all iterations (bold; i.e., the 'DP3' model). All models were tested for beech based on the $LS_{50}$-$LS_{100}$ sample (Sect. 2.4).

[Figure]

**Figure S2. Accuracy of the tested model formulations.** The accuracy was assessed with the Akaike information criterion for small samples (AICc; Eq. S40). The boxes indicate the inner quartile range and the median (middle line). The most extreme values are indicated with dots if outside ±1.5 times the inner quartile range from the $1^{st}$ and $3^{rd}$ quartile, and with whiskers otherwise. Orange dots show the mean, which is further indicated in orange to the right of each box, together with the median indicated in black. The models were labeled as $x_P\_x_A\_x_S\_x_X$, with $x_P$ being the number of leaf development phases (i.e., 2 or 3), $x_A$ being the driver of the aging rate (i.e., A or D for photosynthesis or days, respectively), $x_S$ being the stress rate that is the summed response [i.e., g or h for $g(x)$ or $h(x)$] to the stressors cold (C), shortening (P), dry (D), heat (H), and frost days (F), heavy rain periods (R), and nutrient depletion (N), and $x_X$ indicating the formulation of the senescence rate (i.e., S, P, or X when formulated as a sum, product, or exponential function of aging and stress, respectively). All models were calibrated with the $LS_{50}$-$LS_{100}$ sample (Sect. 2.4).

**Table S5. Senescence summarized across mean annual temperature**

| Calibration | Subject | Variable | 3.8–6.1 °C | 6.1–8.4 °C | 8.4–10.7 °C | 10.7–13.0 °C | 13.0–15.4 °C |
|---|---|---|---|---|---|---|---|
| $LS_{50}$-$LS_{100}$ | Size | Site-years | 329 | 2325 | 3652 | 570 | 48 |
| | Timing | SI [doy] | 172.90 | 164.42 | 156.36 | 151.31 | 149.42 |
| | | $LS_{50}$ [doy] | 291.86 | 291.20 | 288.15 | 282.21 | 270.65 |
| | | $LS_{100}$ [doy] | 290.30 | 290.05 | 291.77 | 296.65 | - |
| | Duration | $LS_{50}$–SI [d] | 118.96 | 126.80 | 131.79 | 130.92 | 121.23 |
| | | $LS_{100}$–SI [d] | 123.90 | 123.94 | 134.81 | 144.23 | - |
| | Cause | Stress | 0.77 | 0.66 | 0.61 | 0.66 | 0.94 |
| | | Aging | 0.21 | 0.33 | 0.38 | 0.33 | 0.04 |
| | | Both | 0.02 | 0.01 | 0.01 | 0.01 | 0.02 |
| | Stressors SI | Cold | 0.46 | 0.52 | 0.41 | 0.28 | 0.15 |
| | | Photoperiod | 0.54 | 0.48 | 0.58 | 0.71 | 0.85 |
| | | Dry | 0.00 | 0.00 | 0.01 | 0.01 | 0.00 |
| | Stressors $LS_{50}$ | Cold | 0.56 | 0.64 | 0.57 | 0.41 | 0.12 |
| | | Photoperiod | 0.44 | 0.36 | 0.42 | 0.59 | 0.88 |
| | | Dry | 0.00 | 0.00 | 0.00 | 0.01 | 0.00 |
| | Stressors $LS_{100}$ | Cold | 0.06 | 0.11 | 0.15 | 0.20 | - |
| | | Photoperiod | 0.94 | 0.89 | 0.85 | 0.80 | - |
| | | Dry | 0.00 | 0.00 | 0.00 | 0.00 | - |
| $LS_{50}$ | Size | Site-years | 334 | 2346 | 3620 | 542 | 45 |
| | Timing | SI [doy] | 132.15 | 124.48 | 116.44 | 111.36 | 109.51 |
| | | $LS_{50}$ [doy] | 282.96 | 283.43 | 283.36 | 282.98 | 282.84 |
| | Duration | $LS_{50}$–SI [d] | 150.81 | 158.95 | 166.92 | 171.62 | 173.33 |
| | Cause | Stress | 0.99 | 1.00 | 0.96 | 0.91 | 0.93 |
| | | Aging | 0.01 | 0.00 | 0.03 | 0.07 | 0.07 |
| | | Both | 0.01 | 0.00 | 0.01 | 0.02 | 0.00 |
| | Stressors SI | Cold | 0.09 | 0.15 | 0.26 | 0.40 | 0.38 |
| | | Photoperiod | 0.91 | 0.85 | 0.74 | 0.60 | 0.62 |
| | | Dry | 0.00 | 0.00 | 0.00 | 0.00 | 0.00 |
| | Stressors $LS_{50}$ | Cold | 0.00 | 0.00 | 0.00 | 0.01 | 0.01 |
| | | Photoperiod | 1.00 | 1.00 | 1.00 | 0.99 | 0.99 |
| | | Dry | 0.00 | 0.00 | 0.00 | 0.00 | 0.00 |

*Note*: The summary is structured in the subjects bin size ('size'), timing, duration, cause, and stressors. Size is given by the count of the evaluated variable site-years. Timing is indicated by the mean day of year [doy] of senescence induction (SI) and of the stages when 50% and 100% of the leaves have turned color or have fallen ($LS_{50}$ and $LS_{100}$, respectively). Duration refers to the periods from SI to $LS_{50}$ and to $LS_{100}$ ($LS_{50}$–SI and $LS_{100}$–SI, respectively) and is given in days [d]. Cause is assessed by the relative number of site-years during which aging versus stress induced senescence (i.e., reached their thresholds first), while the variable both refers to aging and stress reaching their thresholds on the same day. Stressors (i.e., cold stress, photoperiod stress, and dry stress) are compared by their relative contribution to the stress rate that has accumulated by SI, $LS_{50}$, and $LS_{100}$. The underlying model was calibrated with the $LS_{50}$-$LS_{100}$ and $LS_{50}$ samples (Sect. 2.4).

**Table S6. Senescence summarized across mean annual Keetch and Byram drought index**

| Calibration | Subject | Variable | 2.7–23.5 | 23.5–44.2 | 44.2–65.0 | 65.0–85.7 | 85.7–107.0 |
|---|---|---|---|---|---|---|---|
| $LS_{50}$-$LS_{100}$ | Size | Site-years | 6603 | 270 | 45 | 4 | 2 |
| | Timing | SI [doy] | 159.65 | 154.16 | 153.60 | 156.50 | 150.00 |
| | | $LS_{50}$ [doy] | 288.63 | 290.92 | 291.98 | 297.00 | 291.50 |
| | | $LS_{100}$ [doy] | 291.15 | 292.92 | 303.33 | - | - |
| | Duration | $LS_{50}$–SI [d] | 128.98 | 136.73 | 138.34 | 140.50 | 141.50 |
| | | $LS_{100}$–SI [d] | 129.21 | 139.84 | 148.67 | - | - |
| | Cause | Stress | 0.65 | 0.50 | 0.36 | 0.50 | 0.50 |
| | | Aging | 0.34 | 0.49 | 0.64 | 0.50 | 0.50 |
| | | Both | 0.01 | 0.00 | 0.00 | 0.00 | 0.00 |
| | Stressors SI | Cold | 0.43 | 0.56 | 0.74 | 0.79 | 0.50 |
| | | Photoperiod | 0.56 | 0.43 | 0.26 | 0.21 | 0.50 |
| | | Dry | 0.01 | 0.00 | 0.00 | 0.00 | 0.00 |
| | Stressors $LS_{50}$ | Cold | 0.57 | 0.71 | 0.77 | 0.92 | 1.00 |
| | | Photoperiod | 0.43 | 0.29 | 0.23 | 0.08 | 0.00 |
| | | Dry | 0.00 | 0.00 | 0.00 | 0.00 | 0.00 |
| | Stressors $LS_{100}$ | Cold | 0.13 | 0.14 | 0.35 | - | - |
| | | Photoperiod | 0.87 | 0.86 | 0.65 | - | - |
| | | Dry | 0.00 | 0.00 | 0.00 | - | - |
| $LS_{50}$ | Size | Site-years | 6578 | 263 | 39 | 5 | 2 |
| | Timing | SI [doy] | 119.76 | 113.81 | 113.77 | 116.00 | 110.00 |
| | | $LS_{50}$ [doy] | 283.32 | 283.64 | 283.08 | 285.00 | 284.50 |
| | Duration | $LS_{50}$–SI [d] | 163.56 | 169.83 | 169.31 | 169.00 | 174.50 |
| | Cause | Stress | 0.97 | 0.94 | 0.85 | 1.00 | 1.00 |
| | | Aging | 0.02 | 0.04 | 0.13 | 0.00 | 0.00 |
| | | Both | 0.01 | 0.02 | 0.03 | 0.00 | 0.00 |
| | Stressors SI | Cold | 0.22 | 0.32 | 0.22 | 0.04 | 0.61 |
| | | Photoperiod | 0.78 | 0.68 | 0.78 | 0.96 | 0.39 |
| | | Dry | 0.00 | 0.00 | 0.00 | 0.00 | 0.00 |
| | Stressors $LS_{50}$ | Cold | 0.00 | 0.00 | 0.00 | 0.00 | 0.00 |
| | | Photoperiod | 1.00 | 1.00 | 1.00 | 1.00 | 1.00 |
| | | Dry | 0.00 | 0.00 | 0.00 | 0.00 | 0.00 |

*Note*: See Table S5.

**Table S7. Senescence summarized across latitude**

| Calibration | Subject | Variable | 45.8–48.3 °N | 48.3–50.7 °N | 50.7–53.1 °N | 53.1–55.6 °N | 55.6–58 °N |
|---|---|---|---|---|---|---|---|
| LS$_{50}$-LS$_{100}$ | Size | Site-years | 3709 | 1792 | 884 | 512 | 27 |
| | Timing | SI [doy] | 160.66 | 157.82 | 157.32 | 159.14 | 160.96 |
| | | LS$_{50}$ [doy] | 283.32 | 294.36 | 294.96 | 298.69 | 275.42 |
| | | LS$_{100}$ [doy] | 289.14 | 295.61 | 296.53 | 293.06 | 290.43 |
| | Duration | LS$_{50}$–SI [d] | 122.66 | 136.52 | 137.58 | 139.65 | 114.54 |
| | | LS$_{100}$–SI [d] | 124.34 | 141.34 | 141.64 | 135.67 | 130.33 |
| | Cause | Stress | 0.79 | 0.48 | 0.48 | 0.42 | 0.96 |
| | | Aging | 0.21 | 0.51 | 0.51 | 0.57 | 0.04 |
| | | Both | 0.01 | 0.01 | 0.01 | 0.01 | 0.00 |
| | Stressors SI | Cold | 0.30 | 0.64 | 0.58 | 0.64 | 0.13 |
| | | Photoperiod | 0.70 | 0.36 | 0.41 | 0.35 | 0.87 |
| | | Dry | 0.01 | 0.00 | 0.01 | 0.00 | 0.00 |
| | Stressors LS$_{50}$ | Cold | 0.41 | 0.78 | 0.72 | 0.89 | 0.19 |
| | | Photoperiod | 0.59 | 0.21 | 0.28 | 0.11 | 0.81 |
| | | Dry | 0.00 | 0.00 | 0.00 | 0.00 | 0.00 |
| | Stressors LS$_{100}$ | Cold | 0.10 | 0.14 | 0.20 | 0.17 | 0.10 |
| | | Photoperiod | 0.90 | 0.86 | 0.80 | 0.83 | 0.90 |
| | | Dry | 0.00 | 0.00 | 0.00 | 0.00 | 0.00 |
| LS$_{50}$ | Size | Site-years | 3722 | 1739 | 887 | 511 | 28 |
| | Timing | SI [doy] | 120.85 | 117.89 | 117.16 | 119.05 | 120.68 |
| | | LS$_{50}$ [doy] | 283.29 | 283.37 | 283.43 | 283.34 | 283.71 |
| | Duration | LS$_{50}$–SI [d] | 162.44 | 165.47 | 166.27 | 164.29 | 163.04 |
| | Cause | Stress | 0.98 | 0.96 | 0.96 | 0.97 | 1.00 |
| | | Aging | 0.02 | 0.03 | 0.03 | 0.03 | 0.00 |
| | | Both | 0.01 | 0.01 | 0.01 | 0.01 | 0.00 |
| | Stressors SI | Cold | 0.21 | 0.26 | 0.25 | 0.22 | 0.12 |
| | | Photoperiod | 0.79 | 0.74 | 0.75 | 0.78 | 0.88 |
| | | Dry | 0.00 | 0.00 | 0.00 | 0.00 | 0.00 |
| | Stressors LS$_{50}$ | Cold | 0.00 | 0.00 | 0.00 | 0.00 | 0.00 |
| | | Photoperiod | 1.00 | 1.00 | 1.00 | 1.00 | 1.00 |
| | | Dry | 0.00 | 0.00 | 0.00 | 0.00 | 0.00 |

*Note*: See Table S5.

**Table S8. Senescence summarized across elevation**

| Calibration | Subject | Variable | −1–288 m | 288–576 m | 576–864 m | 864–1150 m | 1150–1440 m |
|---|---|---|---|---|---|---|---|
| $LS_{50}$-$LS_{100}$ | Size | Site-years | 2023 | 2767 | 1329 | 666 | 139 |
| | Timing | SI [doy] | 156.43 | 157.27 | 162.20 | 168.12 | 176.01 |
| | | $LS_{50}$ [doy] | 293.58 | 287.55 | 282.78 | 287.59 | 305.68 |
| | | $LS_{100}$ [doy] | 293.29 | 289.58 | 290.45 | 289.03 | - |
| | Duration | $LS_{50}$–SI [d] | 137.14 | 130.28 | 120.62 | 119.47 | 129.67 |
| | | $LS_{100}$–SI [d] | 137.61 | 131.46 | 123.19 | 123.08 | - |
| | Cause | Stress | 0.48 | 0.64 | 0.81 | 0.76 | 0.65 |
| | | Aging | 0.50 | 0.35 | 0.19 | 0.23 | 0.33 |
| | | Both | 0.01 | 0.01 | 0.01 | 0.01 | 0.01 |
| | Stressors SI | Cold | 0.55 | 0.41 | 0.32 | 0.41 | 0.77 |
| | | Photoperiod | 0.44 | 0.59 | 0.67 | 0.58 | 0.23 |
| | | Dry | 0.00 | 0.01 | 0.00 | 0.00 | 0.00 |
| | Stressors $LS_{50}$ | Cold | 0.74 | 0.54 | 0.43 | 0.50 | 0.87 |
| | | Photoperiod | 0.26 | 0.46 | 0.56 | 0.49 | 0.13 |
| | | Dry | 0.00 | 0.00 | 0.00 | 0.00 | 0.00 |
| | Stressors $LS_{100}$ | Cold | 0.16 | 0.13 | 0.12 | 0.06 | - |
| | | Photoperiod | 0.84 | 0.87 | 0.88 | 0.94 | - |
| | | Dry | 0.00 | 0.00 | 0.00 | 0.00 | - |
| $LS_{50}$ | Size | Site-years | 2018 | 2687 | 1368 | 681 | 133 |
| | Timing | SI [doy] | 116.42 | 117.43 | 122.23 | 128.10 | 135.60 |
| | | $LS_{50}$ [doy] | 283.32 | 283.33 | 283.40 | 283.25 | 283.29 |
| | Duration | $LS_{50}$–SI [d] | 166.90 | 165.90 | 161.17 | 155.15 | 147.68 |
| | Cause | Stress | 0.95 | 0.97 | 0.98 | 1.00 | 1.00 |
| | | Aging | 0.04 | 0.02 | 0.01 | 0.00 | 0.00 |
| | | Both | 0.01 | 0.01 | 0.01 | 0.00 | 0.00 |
| | Stressors SI | Cold | 0.28 | 0.25 | 0.17 | 0.12 | 0.05 |
| | | Photoperiod | 0.72 | 0.75 | 0.83 | 0.88 | 0.95 |
| | | Dry | 0.00 | 0.00 | 0.00 | 0.00 | 0.00 |
| | Stressors $LS_{50}$ | Cold | 0.00 | 0.00 | 0.00 | 0.00 | 0.00 |
| | | Photoperiod | 1.00 | 1.00 | 1.00 | 1.00 | 1.00 |
| | | Dry | 0.00 | 0.00 | 0.00 | 0.00 | 0.00 |

*Note*: The bins of elevation are given in m a.s.l.. For further notes see Table S5.

[Figure]

**Figure S3. Date and duration of senescence.** Panel (a) and (b) are based on predictions by the DP3 model calibrated with the  versus LS50  LS100 , respectively (Sect. 2.4). The top row of each panel illustrates the duration of senescence according to LS50 (i.e., the difference in days [d] of the day when 50% of the leaves have turned color or have fallen, LS50, minus the day of senescence induction, SI). The second row of panel (a) shows the duration of senescence according to LS100 (i.e., the difference in days [d] of the day when 100% of the leaves have turned color or have fallen, LS100, minus SI). The third row of panel (a) and the middle row of panel (b) visualize the day of year [doy] of LS50, while the fourth row of panel (a) do so for LS100. The fifth row of panel (a) and the bottom row of panel (b) illustrate the relative amount of cold stress, photoperiod stress, and dry stress that accumulated between SI and LS50. Accordingly, the bottom row of panel (a) shows these relative amounts for the period from SI to LS100. In every row, the x-axis is divided in equally distributed bins among mean annual temperature (MAT, °C), mean annual Keetch and Byram drought index (MAQ), latitude (LAT, °N), and elevation (ELV; m a.s.l.). While the mean and median dates are marked with black dots and grey lines, respectively, the most extreme values are indicated with dots if outside ±1.5 times the inner quartile range from the 1st and 3rd quartile, and with whiskers otherwise.

**S3.2 Model error**

**Table S9. Linear mixed-effects model (LMM) explaining the model error**

| Coefficient | Value | SE | *t* statistic | *p*-value | BF$_{01}$ | Lower 0.5% | Upper 99.5% |
|---|---|---|---|---|---|---|---|
| Intercept | 8.1104 | 1.5297 | 5.3021 | **0.0000** | **0.0000** | 4.1700 | 12.0507 |
| CDD [d] | −1.8144 | 0.2367 | −7.6656 | **0.0000** | **0.0000** | −2.4241 | −1.2047 |
| DM2 [d] | −0.7348 | 0.2367 | −3.1043 | **0.0019** | 0.0414 | −1.3445 | −0.1251 |
| PIA [d] | −0.6290 | 0.2509 | −2.5069 | 0.0122 | 0.1785 | −1.2754 | 0.0173 |
| DP3$_{LS_{50}}$ [d] | 0.1038 | 0.2367 | 0.4385 | 0.6610 | 1.0000 | −0.5059 | 0.7135 |
| δMAT [d °C$^{-1}$] | −2.0028 | 0.1306 | −15.3347 | **0.0000** | **0.0000** | −2.3393 | −1.6664 |
| δMAQ [d] | 0.0901 | 0.0179 | 5.0172 | **0.0000** | **0.0000** | 0.0438 | 0.1363 |
| δ$A_{net}$ [d mol C$^{-1}$ m$^{-2}$] | 0.4148 | 0.0160 | 25.9480 | **0.0000** | **0.0000** | 0.3736 | 0.4560 |
| δLAT [d °$^{-1}$] | 2.1126 | 0.4724 | 4.4725 | **0.0000** | **0.0003** | 0.8959 | 3.3294 |
| δELV [d m$^{-1}$] | 0.0113 | 0.0031 | 3.6690 | **0.0002** | *0.0072* | 0.0034 | 0.0193 |
| SUI [d] | −6.1036 | 1.8142 | −3.3643 | **0.0008** | 0.0193 | −10.7770 | −1.4302 |
| GER [d] | −7.8265 | 2.2319 | −3.5067 | **0.0005** | 0.0124 | −13.5758 | −2.0772 |
| GBR [d] | −24.6277 | 2.9215 | −8.4297 | **0.0000** | **0.0000** | −32.1534 | −17.1020 |
| CDD × δMAT [d °C$^{-1}$] | −0.1457 | 0.1887 | −0.7720 | 0.4401 | 1.0000 | −0.6318 | 0.3404 |
| DM2 × δMAT [d °C$^{-1}$] | −0.1771 | 0.1887 | −0.9386 | 0.3480 | 1.0000 | −0.6632 | 0.3090 |
| PIA × δMAT [d °C$^{-1}$] | 0.0961 | 0.1888 | 0.5092 | 0.6106 | 1.0000 | −0.3901 | 0.5824 |
| DP3$_{LS_{50}}$ × δMAT [d °C$^{-1}$] | −0.0431 | 0.1887 | −0.2283 | 0.8194 | 1.0000 | −0.5292 | 0.4430 |
| CDD × δMAQ [d] | −0.0537 | 0.0276 | −1.9495 | 0.0512 | 0.4806 | −0.1247 | 0.0173 |
| DM2 × δMAQ [d] | −0.0502 | 0.0276 | −1.8225 | 0.0684 | 0.5709 | −0.1212 | 0.0208 |
| PIA × δMAQ [d] | −0.0474 | 0.0281 | −1.6869 | 0.0916 | 0.6704 | −0.1197 | 0.0250 |
| DP3$_{LS_{50}}$ × δMAQ [d] | −0.0147 | 0.0276 | −0.5332 | 0.5939 | 1.0000 | −0.0857 | 0.0563 |
| CDD × δ$A_{net}$ [d mol C$^{-1}$ m$^{-2}$] | 0.0173 | 0.0236 | 0.7323 | 0.4640 | 1.0000 | −0.0435 | 0.0781 |
| DM2 × δ$A_{net}$ [d mol C$^{-1}$ m$^{-2}$] | 0.0093 | 0.0236 | 0.3958 | 0.6922 | 1.0000 | −0.0515 | 0.0702 |
| PIA × δ$A_{net}$ [d mol C$^{-1}$ m$^{-2}$] | 0.0466 | 0.0236 | 1.9739 | 0.0484 | 0.4639 | −0.0142 | 0.1074 |
| DP3$_{LS_{50}}$ × δ$A_{net}$ [d mol C$^{-1}$ m$^{-2}$] | −0.0031 | 0.0236 | −0.1327 | 0.8945 | 1.0000 | −0.0639 | 0.0577 |
| CDD × δLAT [d °$^{-1}$] | −0.0573 | 0.1376 | −0.4162 | 0.6773 | 1.0000 | −0.4117 | 0.2972 |
| DM2 × δLAT [d °$^{-1}$] | −0.0665 | 0.1376 | −0.4830 | 0.6291 | 1.0000 | −0.4209 | 0.2880 |
| PIA × δLAT [d °$^{-1}$] | −0.0344 | 0.1378 | −0.2495 | 0.8030 | 1.0000 | −0.3892 | 0.3205 |
| DP3$_{LS_{50}}$ × δLAT [d °$^{-1}$] | 0.0172 | 0.1376 | 0.1252 | 0.9004 | 1.0000 | −0.3372 | 0.3716 |
| CDD × δELV [d m$^{-1}$] | −0.0012 | 0.0013 | −0.9438 | 0.3453 | 1.0000 | −0.0045 | 0.0021 |
| DM2 × δELV [d m$^{-1}$] | −0.0011 | 0.0013 | −0.8856 | 0.3759 | 1.0000 | −0.0044 | 0.0021 |
| PIA × δELV [d m$^{-1}$] | −0.0014 | 0.0013 | −1.0789 | 0.2806 | 0.9940 | −0.0046 | 0.0019 |
| DP3$_{LS_{50}}$ × δELV [d m$^{-1}$] | 0.0000 | 0.0013 | 0.0139 | 0.9889 | 1.0000 | −0.0032 | 0.0033 |

*Note*: The LMM was fitted to the response variable 'model error' [i.e., $x_{s,i} - x_{o,i}$, the difference in days calculated as the predicted minus the observed date for each stage and site-year (*i*)] in the validation sample (Sect. 2.6 and S2.3), based on 41 068 observations, and resulted in an adjusted R$^2$ of 0.44 and a proportion of the deviance explained of 0.44. The random intercepts were grouped by site with $\sigma_b$ = 9.32 d (99% confidence interval 8.26 ≤ $\sigma_b$ ≤ 10.52 d). SE is the standard error, while 'Lower 0.05%' and 'Upper 99.5%' indicate the lower and upper boundaries of the 99% confidence interval. Bold *p*-values are indicate significant fixed effects at α = 0.01 (i.e., $p$ ≤ 0.005 for a two-sided hypothesis test), bold and italic minimum Bayes factors (BF$_{01}$) indicate decisive and very strong fixed effects (i.e., *BF$_{01}$* ≤ 1/1000 and *BF$_{01}$* ≤ 1/100, respectively). The intercept represents the base line, i.e., the model error according to the Null model for the reference level Austria. CDD, DM2, PIA, and DP3 are the factorized models, while SUI, GER, and GBR are the factorized countries Switzerland, Germany, and Great Britain, respectively. The random intercepts were grouped by 'site'. All models ware calibrated and validated with the LS$_{50}$ sample (Sect. 2.4).

**Table S10. Interacting effects according to the LMM**

| Variable | Model | Country | Estimate | SE | 0.5 % | 99.5 % | Equation |
|---|---|---|---|---|---|---|---|
| Country [d] | Null | AUT | 8.11 | 1.53 | 4.17 | 12.05 | $\beta_0$ |
| | | SUI | 2.01 | 1.50 | −1.86 | 5.87 | $\beta_0 + \text{SUI}$ |
| | | GER | 0.28 | 1.34 | −3.16 | 3.73 | $\beta_0 + \text{GER}$ |
| | | GBR | −16.52 | 2.02 | −21.71 | −11.32 | $\beta_0 + \text{GBR}$ |
| | CDD | AUT | 6.30 | 1.55 | 2.31 | 10.28 | $\beta_0 + \text{CDD}$ |
| | | SUI | 0.19 | 1.52 | −3.72 | 4.11 | $\beta_0 + \text{CDD} + \text{SUI}$ |
| | | GER | −1.53 | 1.33 | −4.96 | 1.90 | $\beta_0 + \text{CDD} + \text{GER}$ |
| | | GBR | −18.33 | 2.00 | −23.49 | −13.17 | $\beta_0 + \text{CDD} + \text{GBR}$ |
| | DM2 | AUT | 7.38 | 1.55 | 3.39 | 11.36 | $\beta_0 + \text{DM2}$ |
| | | SUI | 1.27 | 1.52 | −2.64 | 5.18 | $\beta_0 + \text{DM2} + \text{SUI}$ |
| | | GER | −0.45 | 1.33 | −3.88 | 2.98 | $\beta_0 + \text{DM2} + \text{GER}$ |
| | | GBR | −17.25 | 2.00 | −22.41 | −12.09 | $\beta_0 + \text{DM2} + \text{GBR}$ |
| | PIA | AUT | 7.48 | 1.52 | 3.56 | 11.41 | $\beta_0 + \text{PIA}$ |
| | | SUI | 1.38 | 1.49 | −2.47 | 5.23 | $\beta_0 + \text{PIA} + \text{SUI}$ |
| | | GER | −0.35 | 1.36 | −3.84 | 3.15 | $\beta_0 + \text{PIA} + \text{GER}$ |
| | | GBR | −17.15 | 2.04 | −22.40 | −11.89 | $\beta_0 + \text{PIA} + \text{GBR}$ |
| | DP3$_{\text{LS}_{50}}$ | AUT | 8.21 | 1.55 | 4.23 | 12.20 | $\beta_0 + \text{DP3}_{\text{LS}_{50}}$ |
| | | SUI | 2.11 | 1.52 | −1.80 | 6.02 | $\beta_0 + \text{DP3}_{\text{LS}_{50}} + \text{SUI}$ |
| | | GER | 0.39 | 1.33 | −3.05 | 3.82 | $\beta_0 + \text{DP3}_{\text{LS}_{50}} + \text{GER}$ |
| | | GBR | −16.41 | 2.00 | −21.57 | −11.25 | $\beta_0 + \text{DP3}_{\text{LS}_{50}} + \text{GBR}$ |
| $\delta$ELV [d 100 m$^{-1}$] | Null | AC | 1.13 | 0.31 | 0.34 | 1.93 | $100\,\delta\text{ELV}$ |
| | CDD | | 1.01 | 0.32 | 0.20 | 1.83 | $100\,(\delta\text{ELV} + \text{CDD} \times \delta\text{ELV})$ |
| | DM2 | | 1.02 | 0.32 | 0.20 | 1.84 | $100\,(\delta\text{ELV} + \text{DM2} \times \delta\text{ELV})$ |
| | PIA | | 1.00 | 0.32 | 0.18 | 1.82 | $100\,(\delta\text{ELV} + \text{PIA} \times \delta\text{ELV})$ |
| | DP3$_{\text{LS}_{50}}$ | | 1.14 | 0.32 | 0.32 | 1.95 | $100\,(\delta\text{ELV} + \text{DP3}_{\text{LS}_{50}} \times \delta\text{ELV})$ |
| $\delta$LAT [d °N$^{-1}$] | Null | | 2.11 | 0.47 | 0.90 | 3.33 | $\delta\text{LAT}$ |
| | CDD | | 2.06 | 0.48 | 0.82 | 3.29 | $\delta\text{LAT} + \text{CDD} \times \delta\text{LAT}$ |
| | DM2 | | 2.05 | 0.48 | 0.81 | 3.28 | $\delta\text{LAT} + \text{DM2} \times \delta\text{LAT}$ |
| | PIA | | 2.08 | 0.48 | 0.84 | 3.31 | $\delta\text{LAT} + \text{PIA} \times \delta\text{LAT}$ |
| | DP3$_{\text{LS}_{50}}$ | | 2.13 | 0.48 | 0.90 | 3.36 | $\delta\text{LAT} + \text{DP3}_{\text{LS}_{50}} \times \delta\text{LAT}$ |
| $\delta$MAQ [d 100$^{-1}$] | Null | | 9.01 | 1.79 | 4.38 | 13.63 | $100\,\delta\text{MAQ}$ |
| | CDD | | 3.63 | 2.29 | −2.28 | 9.54 | $100\,(\delta\text{MAQ} + \text{CDD} \times \delta\text{MAQ})$ |
| | DM2 | | 3.98 | 2.29 | −1.93 | 9.89 | $100\,(\delta\text{MAQ} + \text{DM2} \times \delta\text{MAQ})$ |
| | PIA | | 4.27 | 2.37 | −1.83 | 10.37 | $100\,(\delta\text{MAQ} + \text{PIA} \times \delta\text{MAQ})$ |
| | DP3$_{\text{LS}_{50}}$ | | 7.54 | 2.29 | 1.63 | 13.45 | $100\,(\delta\text{MAQ} + \text{DP3}_{\text{LS}_{50}} \times \delta\text{MAQ})$ |
| $\delta$MAT [d 10°C$^{-1}$] | Null | | −20.03 | 1.31 | −23.39 | −16.66 | $10\,\delta\text{MAT}$ |
| | CDD | | −21.49 | 1.69 | −25.84 | −17.13 | $10\,(\delta\text{MAT} + \text{CDD} \times \delta\text{MAT})$ |
| | DM2 | | −21.80 | 1.69 | −26.16 | −17.44 | $10\,(\delta\text{MAT} + \text{DM2} \times \delta\text{MAT})$ |
| | PIA | | −19.07 | 1.69 | −23.43 | −14.71 | $10\,(\delta\text{MAT} + \text{PIA} \times \delta\text{MAT})$ |
| | DP3$_{\text{LS}_{50}}$ | | −20.46 | 1.69 | −24.82 | −16.10 | $10\,(\delta\text{MAT} + \text{DP3}_{\text{LS}_{50}} \times \delta\text{MAT})$ |
| $\delta A_{\text{net}}$ [d 10 mol C$^{-1}$ m$^{-2}$] | Null | | 4.15 | 0.16 | 3.74 | 4.56 | $10\,\delta A_{\text{net}}$ |
| | CDD | | 4.32 | 0.21 | 3.78 | 4.86 | $10\,(\delta A_{\text{net}} + \text{CDD} \times \delta A_{\text{net}})$ |
| | DM2 | | 4.24 | 0.21 | 3.70 | 4.78 | $10\,(\delta A_{\text{net}} + \text{DM2} \times \delta A_{\text{net}})$ |
| | PIA | | 4.61 | 0.21 | 4.08 | 5.15 | $10\,(\delta A_{\text{net}} + \text{PIA} \times \delta A_{\text{net}})$ |
| | DP3$_{\text{LS}_{50}}$ | | 4.12 | 0.21 | 3.58 | 4.65 | $10\,(\delta A_{\text{net}} + \text{DP3}_{\text{LS}_{50}} \times \delta A_{\text{net}})$ |

*Note*: The interacting effects of the LMM (Table S9) were calculated with the Delta method (Chpt. 5.1.4 in Fox and Weisberg, 2019; Chpt. 9.9 in Wasserman, 2004) according to the displayed equation, together with their standard error (SE) and 99% confidence interval (i.e., the 0.5% lower bound and 99.5% upper bound). AUT, SUI, GER, and GBR refer to the countries Austria, Switzerland, Germany, and Great Britain, respectively, while AC marks estimates across countries. The unit for $\delta A_{\text{net}}$ is d 10 mol C$^{-1}$ m$^{-2}$.

[revised manuscript text omitted]